# Analytical Probability Distributions and Exact Expectation-Maximization for Deep Generative Networks

**Randall Balestriero**
ECE Department
Rice University

**Sébastien Paris**
Univ Toulon, Aix Marseille Univ,
CNRS, LIS, Toulon, France

**Richard G. Baraniuk**
ECE Department
Rice University

## Abstract

Deep Generative Networks (DGNs) with probabilistic modeling of their output and latent space are currently trained via Variational Autoencoders (VAEs). In the absence of a known analytical form for the posterior and likelihood expectation, VAEs resort to approximations, including (Amortized) Variational Inference (AVI) and Monte-Carlo sampling. We exploit the Continuous Piecewise Affine property of modern DGNs to derive their posterior and marginal distributions as well as the latter's first two moments. These findings enable us to derive an analytical Expectation-Maximization (EM) algorithm for gradient-free DGN learning. We demonstrate empirically that EM training of DGNs produces greater likelihood than VAE training. Our new framework will guide the design of new VAE AVI that better approximates the true posterior and open new avenues to apply standard statistical tools for model comparison, anomaly detection, and missing data imputation.

## 1 Introduction

Deep Generative Networks (DGNs), which map a low-dimensional latent variable $z$ to a higher-dimensional generated sample $x$ are the state-of-the-art methods for a range of machine learning applications, including anomaly detection, data generation, likelihood estimation, and exploratory analysis across a wide variety of datasets [1–4].

Training of DGNs roughly falls into two camps: (i) By leveraging an adversarial network as in a Generative Adversarial Network (GAN) [5] to turn the method into an adversarial game; and (ii) by modeling the latent variable and observed variables as random variables and performing some flavor of likelihood maximization training. A widely used solution to likelihood based DGN training is via a Variational Autoencoder (VAE) [6]. The popularity of the VAE is due to its intuitive and interpretable loss function, which is obtained from likelihood estimation, and its ability to exploit standard estimation techniques ported from the probabilistic graphical models literature.

Yet, VAEs offer only an *approximate* solution for likelihood based training of DGNs. In fact, all current VAEs employ three major approximation steps in the likelihood maximization process. First, the true (unknown) posterior is approximated by a variational distribution. This estimate is governed by some free parameters that must be optimized to fit the variational distribution to the true posterior. VAEs estimate such parameters by means of an alternative network, the *encoder*, with the datum as input and the predicted optimal parameters as output. This step is referred to as Amortized Variational Inference (AVI), as it removes the explicit, per datum, optimization by a single deep network (DN) pass. Second, as in any latent variable model, the complete likelihood is estimated by a lower bound (ELBO) obtained from the expectation of the likelihood taken under the posterior or variational distribution. With a DGN, this expectation is unknown, and thus VAEs estimate the ELBO by Monte-Carlo (MC) sampling. Third, the maximization of the MC-estimated ELBO, which

drives the parameters of the encoder to better model the data distribution and the encoder to produce better variational parameter estimates, is performed by some flavor of gradient descend (GD).

These VAE approximation steps enable rapid training and test-time inference of DGNs. However, due to the lack of analytical forms for the posterior, ELBO, and explicit (gradient free) parameter updates, it is not possible to measure the above steps' quality or effectively improve them. Since the true posterior and expectation are unknown, current VAE research roughly fall into three camps: (i) developing new and more complex output and latent distributions [7, 8], such as the truncated distribution; (ii) improving the various estimation steps by introducing complex MC sampling with importance re-weighted sampling [9]; (iii) providing different estimates of the posterior with moment matching techniques [10, 11]. More recently, [12] exploited the special continuous piecewise affine structure of current ReLU DGNs to develop an approximation of the posterior distribution based on mode estimation and DGN linearization leading to Laplacian VAEs. Nevertheless, derivation of analytical DGN distributions was not considered.

***In this paper, we advance both the theory and practice of DGNs and VAEs by computing the exact analytical posterior and marginal distributions of any DGN employing continuous piecewise affine (CPA) nonlinearities. The knowledge of these distributions enables us to perform exact inference without resorting to AVI or MC-sampling and to train the DGN in a gradient-free manner with guaranteed convergence.***

The analytical distributions we obtain provide first-of-their-kind insights into (i) how DGNs model the data distributions à la Mixture of Probabilistic Principal Component Analysis (MPPCA), (ii) how inference is performed and is akin Generative Latent Optimization models [13], (iii) the roles of each DGN parameter and how are they updated, and (iv) the impact of DGN architecture and regularization choice in the form of the DGN distributions and layer weights. The exact likelihood and marginal computation also enable the use of standard statistical model comparison tools such as the Akkaike Information Criterion (AIC) [14] and Bayesian Information Criterion (BIC) [15] and inspire new, more reliable anomaly detection approaches.

Access to the exact posterior also enables us to quantify the approximation error of the AVI and MC sampling of VAEs and guide the development of VAEs by leveraging the analytical posterior to design more adapted variational distributions. In fact, current VAEs suffer from occasional training instabilities [16, 17]; we validate the empirical observation that training instabilities emerge from an inadequate variational estimation of the posterior.

We summarize our main contributions as follows:

**[C1]** We leverage the CPA property of current DGNs to obtain the analytical form of their conditional, marginal, and posterior distributions, which are mixtures of truncated Gaussians and relate DGN density modeling to MPPCA and MFA (Sec. 3.1). We develop new algorithms and methods to compute the DGN latent space partition, per-region affine mappings, and per-region Gaussian integration (Sec. 3.2).

**[C2]** We leverage the analytical form of a DGN's posterior distribution to obtain its first two moments. We then leverage these moments to obtain the analytical expectation of the complete likelihood with respect to the DGN posterior (E-step), which enables *encoder-free EM training with guaranteed convergence* (Sec. 4.1). We also derive the analytical M-step, which enables for the first time *guaranteed and rapid gradient-free learning of DGNs* (Sec. 4.2). The analytical E-step allows us to interpret how the expected latent representation of an input is formed, while the M-step demonstrates how information is propagated through the layers akin to the backpropagation encountered in gradient descent.

**[C3]** We compare our exact E-step to standard VAE training to demonstrate that the VAE inference step is to blame for unstable training. We also demonstrate that EM-based DGN training provides much faster and more stable convergence (Sec. 4.3) and suggest new directions to leverage the analytical distributions to improve VAE models.

Reproducible code for all experiments and figures are available on Github at https://github.com/RandallBalestriero/EMDGN.git. The proofs of all results are provided in the Supplementary Material.

## 2   Background

**Max-Affine Spline Deep Generative Networks.** A deep generative network (DGN) is an operator $g$ that maps a (typically low-dimensional) latent vector $\boldsymbol{z} \in \mathbb{R}^S$ to an observation $\boldsymbol{x} \in \mathbb{R}^D$ [1] by composing $L$ intermediate *layer* mappings $g^\ell, \ell = 1, \ldots, L$, that combine affine operators such as the *fully connected operator* (simply an affine transformation defined by weight matrix $\boldsymbol{W}^\ell$ and bias vector $\boldsymbol{v}^\ell$), *convolution operator* (with circulent $\boldsymbol{W}^\ell$), and, nonlinear operators such as the *activation operator* (applying a scalar nonlinearity such as the ubiquitous ReLU), or the *(max-)upsampling operator*. Precise definitions of these operators can be found in [18].

In this paper, we focus on DGNs employing arbitrary affine operators and continuous piecewise affine (CPA) nonlinearities, such as the ReLU, leaky-ReLU, and absolute value activations, and spatial/channel max-pooling. In this case, the entire DGN is the composition of *Max-Affine Spline Operators* (MASOs) [19] and is overall a CPA operator [20–23]. As such, DGNs inherit a latent space partition $\Omega$ of $\mathbb{R}^S$ and a per-region affine mapping

$$g(\boldsymbol{z}) = \boldsymbol{A}_\omega \boldsymbol{z} + \boldsymbol{b}_\omega, \forall \omega \in \Omega, \tag{1}$$

where the per-region slope and bias parameters are a function of the per-layer parameters $\boldsymbol{W}_\ell, \boldsymbol{b}_\ell$. For various properties of such CPA DGNs, see [24], and for details on the partition, see [25]. In this paper we will make explicit the per-region affine mappings; to this end, it is practical to encode the derivatives of the DGN nonlinearities in the matrices $\boldsymbol{D}_\ell$. For activation operators, this is a square diagonal matrix with values $\in \{\eta, 1\}$ ($\eta > 0$ for leaky-ReLU, $\eta = 0$ for ReLU, and $\eta = -1$ for absolute value). For the max-pooling operator, it is a rectangular matrix filled with $\{0, 1\}$ values based on the pooling $\arg\max$. We thus obtain

$$\boldsymbol{A}_\omega = \boldsymbol{W}^L \boldsymbol{D}_\omega^{L-1} \boldsymbol{W}^{L-1} \ldots \boldsymbol{D}_\omega^1 \boldsymbol{W}^1 \quad \text{and} \quad \boldsymbol{b}_\omega = \boldsymbol{v}^L + \sum_{i=1}^{L-1} \boldsymbol{W}^L \boldsymbol{D}_\omega^{L-1} \boldsymbol{W}^{L-1} \ldots \boldsymbol{D}_\omega^i \boldsymbol{v}^i. \tag{2}$$

Throughout the rest of the paper, the upper index will indicate the layer and not a power.

**Variational Expectation-Maximization.** A Probabilistic Graphical Model (PGM) combines probability and graph theory into an organized data structure that expresses the relationships between a collection of random variables: the *observed* variables collected into $\boldsymbol{x}$ and the *latent*, or unobserved, variables collected into $\boldsymbol{z}$ [26]. The parameters $\theta$ that govern the PGM probability distributions are learned from observations $\boldsymbol{x}_i \sim \boldsymbol{x}, i = 1, \ldots, N$, requiring estimation of the unobserved $\boldsymbol{z}_i, \forall i$. This inference-optimization is commonly done with the Expectation-Maximization (EM) algorithm [27].

The EM algorithm consists of (i) estimating each $\boldsymbol{z}_i$ from the Expectation of the complete log-density taken with respect to the posterior distribution under the current parameters at time $t$; (ii) Maximizing the estimated complete log-likelihood to produce the updated parameters $\theta_{t+1}$. The estimated complete log-likelihood obtained from the E-step is a tight lower bound to the true complete log-likelihood; this lower bound is maximized in the M-step. This process has many attractive theoretical properties, including guaranteed convergence to a local minimum of the likelihood [28].

In the absence of closed form or tractable posterior, an alternative (non-tight) lower bound can be obtained by using a *variational distribution* instead. This distribution is governed by parameters $\gamma$ that are optimized to make this distribution as close as possible to the true posterior. This process is results in a *variational* E (VE) step [29] or *variational inference* (VI). The *tightness* of the lower bound is measured by the Kullback–Leibler (KL) divergence between the variational and true posterior distributions. Minimization of this divergence cannot be done directly (due to the absence of tractable posterior) but rather indirectly by maximizing the so-called evidence lower bound (ELBO) via

$$\log(p(\boldsymbol{x})) = \underbrace{\mathbb{E}_{q(\boldsymbol{z}|\gamma)}[\log(p(\boldsymbol{x}, \boldsymbol{z}|\theta))] + \mathrm{H}(q(\boldsymbol{z}|\gamma))}_{\text{ELBO}} + \mathrm{KL}(q(\boldsymbol{z}|\gamma)\|p(\boldsymbol{z}|\boldsymbol{x}, \theta)), \tag{3}$$

with $q$ the variational distribution and $H$ the (differential) entropy. Maximization the ELBO with respect to $\gamma$ produces the $\gamma^*$ that adapts $q(\boldsymbol{z}|\gamma^*)$ to fit as closely as possible to the true posterior. Finally, maximizing the ELBO with respect to the PGM parameters $\theta$ provides $\theta_{t+1}$; this can be performed on the entire dataset or on mini-batches [30].

**Variational AutoEncoders.** A *Variational AutoEncoder* (VAE) uses a minimal *probabilistic graphical model* (PGM) with just a few nodes but highly nonlinear inter-node relations [31, 32]. The

use of DNs to model the nonlinear relations originated in [33–35] and has been born again with VAEs [6]. Many variants have been developed, but the core approach consists of modeling the latent distribution over $z$ with a Gaussian or uniform distribution and then modeling the data distribution as $x = g(z) + \epsilon$ with $\epsilon$ some noise distribution and $g$ a DGN. Learning the DGN/PGM parameters requires inference of the latent variables $z$. This inference is performed in VAEs by producing an *amortized* VI where a second *encoder* DN $f$ produces $\gamma_n^* = f(x_n)$ from (3). Hence, the encoder is fed with an observation $x$ and outputs its estimate of the optimal variational parameters that minimizes the KL divergence between the variational distribution and true posterior. During learning, the encoder adapts to make better estimates $f(x_n)$ of the optimum parameters $\gamma_n$. Then, the ELBO is estimated with some flavor of Monte-Carlo (MC) sampling (since its analytical form is not known), and the maximization of the $\theta$ parameters is solved iteratively using some flavor of gradient descent.

## 3 Posterior and Marginal Distributions of Deep Generative Networks

We now derive analytical forms of the key DGN distributions by exploiting the CPA property. In Sec. 4 we will use this result to derive the EM learning algorithm for DGNs and study the VAE inference approximation versus the analytical one.

Our key insight is that a CPA DGN consists of an implicit latent space partition and an associated per-region affine mapping (recall (1)). In a DGN, propagating a latent datum $z$ through the layers progressively builds the $A_\omega, b_\omega$. We now demonstrate that turning this region selection process explicit, the analytical DGN marginal and posterior distributions can be obtained.

### 3.1 Conditional, Marginal and Posterior Distributions of Deep Generative Networks

Throughout the sequel we will consider the commonly employed case of a centered Gaussian latent prior and centered Gaussian noise [36] as

$$p(x|z) = \phi(x; g(z), \Sigma_x), \ p(z) = \phi(z; 0, \Sigma_z), \tag{4}$$

with $\phi$ the multivariate Gaussian density function with given mean and covariance matrix [37]. When using CPA DGNs, the generator mapping is continuous and piecewise affine with an underlying latent space partition and per-region mapping as in (1). We can thus obtain the analytical form of the conditional distribution of $x$ given the latent vector $z$ as follows.

**Lemma 1.** *The DGN conditional distribution is given by* $p(x|z) = \sum_{\omega \in \Omega} \mathbb{1}_{z \in \omega} \phi(x; A_\omega z + b_\omega, \Sigma_x)$ *with per-region parameters from (2).*

This type of data modeling is closely related to MPPCA [38] that combines multiple PPCAs [39] and MFA [40,41] that combines multiple factor analyzers [42]. The associated PGMs represent the data distribution with $R$ components and leverage an explicit categorical distribution $t \sim \text{Cat}(\pi)$, leading to the conditional input distributions $x|(z,t) = \sum_{r=1}^R \mathbb{1}_{r=t}(W_r z + v_r) + \epsilon$, with $W_r, v_r$ denoting the per-component affine parameters and with $\Sigma_x$ diagonal (MPPCA) or fully occupied (MFA) and $z \sim \mathcal{N}(\mu_z, \Sigma_z)$. Note, however, that neither MPPCA nor MFA impose continuity in the $(t, z) \mapsto x$ mapping as opposed to a DGN. To formalize this, consider an (arbitrary) ordering of the DGN latent space regions as $\omega_1, \dots, \omega_R$ with $R = \text{Card}(\Omega)$. We also denote by $\Phi_\omega$ the cumulative density function on $\omega_r$ (integral of the density function on $\omega_r$).

**Proposition 1.** *A DGN with distributions given by (4) corresponds to a continuous MPPCA (or MFA) model with implicit categorical variable given by $p(t = r) = \Phi_{\omega_r}(0, \Sigma_z), W_r = A_{\omega_r}, v_r = b_{\omega_r}, R = \text{Card}(\Omega)$ and $\Sigma_x = \sigma I$ (or full $\Sigma_x$).*

Note that this result generalizes the results of [12,43], which showed that shallow DGNs and deep linear DGNs fall back to a PPCA model. This can be easily seen from the formula in Lemma 1 by setting the DGN $g$ to be linear as in $g(z) = Wz + b + \epsilon$; in that case, the partition is only made of a single region (the entire DGN input space), and the (single) affine parameters are $A_\omega = W, b_\omega = b$.

We now calculate the marginal $p(x)$ and posterior $p(z|x)$ distributions. The former will be of use to compute the likelihood, while the latter will enable us to derive the analytical E-step in the next section.

**Theorem 1.** *The marginal and posterior distributions of a CPA DGN are given by*

$$p(x) = \sum_{\omega \in \Omega} \phi(x; b_\omega, \Sigma_x + A_\omega \Sigma_z A_\omega^T) \Phi_\omega(\mu_\omega(x), \Sigma_\omega), \tag{5}$$

$$p(\boldsymbol{z}|\boldsymbol{x}) = p(\boldsymbol{x})^{-1} \sum_{\omega \in \Omega} \mathbb{1}_{\boldsymbol{z} \in \omega} \phi(\boldsymbol{x}; \boldsymbol{b}_\omega, \boldsymbol{\Sigma}_{\boldsymbol{x}} + \boldsymbol{A}_\omega \boldsymbol{\Sigma}_{\boldsymbol{z}} \boldsymbol{A}_\omega^T) \phi(\boldsymbol{z}; \boldsymbol{\mu}_\omega(\boldsymbol{x}), \boldsymbol{\Sigma}_\omega), \qquad (6)$$

$$\text{with } \boldsymbol{\mu}_\omega(\boldsymbol{x}) = \boldsymbol{\Sigma}_\omega \left( \boldsymbol{A}_\omega^T \boldsymbol{\Sigma}_{\boldsymbol{x}}^{-1}(\boldsymbol{x} - \boldsymbol{b}_\omega) \right), \text{ and } \boldsymbol{\Sigma}_\omega = \left( \boldsymbol{\Sigma}_{\boldsymbol{z}}^{-1} + \boldsymbol{A}_\omega^T \boldsymbol{\Sigma}_{\boldsymbol{x}}^{-1} \boldsymbol{A}_\omega \right)^{-1}. \qquad (7)$$

We demonstrate how to compute the integral of a multivariate Gaussian on a polytopal domain $(\Phi_\omega(\boldsymbol{\mu}_\omega(\boldsymbol{x}), \boldsymbol{\Sigma}_\omega))$ in the next section. Note that for both the marginal and the posterior distribution, when considering a specific region $\omega \in \Omega$, those distributions are parametrized by a region-specific mean $\boldsymbol{\mu}_\omega(\boldsymbol{x})$ and covariance $\boldsymbol{\Sigma}_\omega$ that we can interpret. For that purpose, consider $\boldsymbol{\Sigma}_{\boldsymbol{x}} = I, \boldsymbol{\Sigma}_{\boldsymbol{z}} = I$ to obtain $\boldsymbol{\mu}_\omega(\boldsymbol{x}) = (I + \boldsymbol{A}_\omega^T \boldsymbol{A}_\omega)^{-1} \boldsymbol{A}_\omega^T (\boldsymbol{x} - \boldsymbol{b}_\omega)$. That is, the bias of the per-region affine mapping is removed from the input which is then mapped back to the latent space via $\boldsymbol{A}_\omega^T$ and whitened by the "regularized" inverse of the correlation matrix of $\boldsymbol{A}_\omega$. Note that $\boldsymbol{A}_\omega^T$ backpropagates the signal from the output to the latent space in the same way that gradients are backpropagated during gradient learning in a DN. We further highlight that the specific form of the posterior is a mixture of truncated Gaussians [44], a truncated Gaussian being a Gaussian distribution for which the domain $\mathbb{R}^S$ has been constrained to a (convex) sub-domain, $\omega$ in our case.

**Proposition 2.** *The DGN posterior distribution is a mixture of* $\mathrm{Card}(\Omega)$ *truncated Gaussians, each truncated on a different polytope $\omega \in \Omega$ with mean $\boldsymbol{\mu}_\omega(\boldsymbol{x})$ and covariance $\boldsymbol{\Sigma}_\omega$ from (7).*

The above proposition is crucial to designing better VAEs. In fact, recalling Sec. 2, VAEs approximate the DGN posterior with a user defined variational distribution. In most of practical cases, this is taken to be a unimodal (Gaussian) distribution. However, as per the analytical posterior that we obtain, unimodal variational distribution cannot capture the multimodality of the true posterior, leading to a poor variational EM step. Based on our result, practitioners should thus favor as much as possible multimodal variational distributions, for instance by employing a mixture of Gaussians for $q(\boldsymbol{z}|\gamma)$ (recall (3) as in [45]. We now propose a specific study in the zero-noise limit.

**Zero-Noise Limit and Generative Latent Optimization (GLO) Models.** In the zero-noise limit ($\boldsymbol{\Sigma}_{\boldsymbol{x}} = \sigma I$ and $\sigma \to 0$), the posterior takes a very special form that we can analyze. Denote by $\boldsymbol{z}^*(\boldsymbol{x}) \triangleq \arg\min_{\boldsymbol{z}} \|\boldsymbol{x} - g(\boldsymbol{z})\|_2^2 + \boldsymbol{z}^T \boldsymbol{\Sigma}_{\boldsymbol{z}} \boldsymbol{z}$, the (regularized) latent vector that produces the closest output from an observation $\boldsymbol{x}$. That is, the vector in the DGN input space that provides the best approximation of $\boldsymbol{x}$ while being regularized based on $\boldsymbol{\Sigma}_{\boldsymbol{z}}$.

**Lemma 2.** *In the zero-noise limit, the DGN posterior distribution converges to a Dirac distribution positioned in the $\boldsymbol{z}$-space at $\boldsymbol{z}^*(\boldsymbol{x})$ as $\lim_{\sigma \to 0} p(\boldsymbol{z}|\boldsymbol{x}) = \delta(\boldsymbol{z} - \boldsymbol{z}^*(\boldsymbol{x}))$.*

Interestingly, the GLO model [13] performs DGN training by first inferring a latent vector akin to $\boldsymbol{z}^*(\boldsymbol{x})$ but without the $\ell_2$ regularization $\boldsymbol{z}^T \boldsymbol{\Sigma}_{\boldsymbol{z}} \boldsymbol{z}$. Instead, GLO and its extensions employ clipping of $\boldsymbol{z}$ which corresponds to a Uniform distribution of $\boldsymbol{z}$ (instead of the Gaussian distribution).

**Proposition 3.** *The GLO-inferred DGN latent variable associated to an observation $\boldsymbol{x}$ corresponds to the maximum a posteriori estimate of the zero-noise limit posterior distribution and with uninformative prior (large $\boldsymbol{\Sigma}_{\boldsymbol{z}}$) or with uniform prior $\boldsymbol{z} \sim \mathcal{U}([a,b])$ when using $[a,b]$ clipping).*

### 3.2 Gaussian Integration on the Deep Generative Network Latent Partition

We now turn to the computation of the DGN marginal (5) and posterior (6) distributions, for which we need to integrate over all of the regions $\omega \in \Omega$ in the latent space partition.

**Obtaining the DGN Partition.** Each region $\omega \in \Omega$ is a polytope that can be explicitly described via a system of inequalities involving the up-to-layer $\ell$ mappings

$$\boldsymbol{A}_\omega^{1 \to \ell} \triangleq \boldsymbol{W}^\ell \boldsymbol{D}_\omega^{\ell-1} \boldsymbol{W}^{\ell-1} \dots \boldsymbol{D}_\omega^1 \boldsymbol{W}^1 \text{ and } \boldsymbol{b}_\omega^{1 \to \ell} \triangleq \boldsymbol{v}^\ell + \sum_{i=1}^{\ell-1} \boldsymbol{W}^\ell \boldsymbol{D}_\omega^{\ell-1} \boldsymbol{W}^{\ell-1} \dots \boldsymbol{D}_\omega^i \boldsymbol{v}^i, \quad (8)$$

producing the pre-activation feature maps $\boldsymbol{h}^\ell(\boldsymbol{z}) \in \mathbb{R}^{D^\ell}$ by $\boldsymbol{h}^\ell(\boldsymbol{z}) = \boldsymbol{A}_\omega^{1 \to \ell} \boldsymbol{z} + \boldsymbol{b}_\omega^{1 \to \ell}$ and with $\boldsymbol{A}_\omega^{1 \to \ell} \in \mathbb{R}^{D^\ell \times S}$ and $\boldsymbol{b}_\omega^{1 \to \ell} \in \mathbb{R}^{D^\ell}$. Note that we have, in particular, that $\boldsymbol{A}_\omega^L = \boldsymbol{A}_\omega$ and $\boldsymbol{b}_\omega^L = \boldsymbol{b}_\omega$ from (2). When using standard activation functions such as (leaky-)ReLU or absolute value, the sign of the pre-activation defines the activation state; denote this by $\boldsymbol{q}^\ell = \mathrm{sign}(\boldsymbol{h}^\ell(\boldsymbol{z}))$ and collect all of the per-layer signs into $\boldsymbol{q}$. Without degenerate weights, the sign patterns produced by $\boldsymbol{q}(\boldsymbol{z}), \forall \boldsymbol{z}$ are tied to the regions $\omega \in \Omega$; we will thus use interchangeably $\boldsymbol{q}(\boldsymbol{z})$ with $\boldsymbol{z} \in \omega$ and $\boldsymbol{q}(\omega)$.

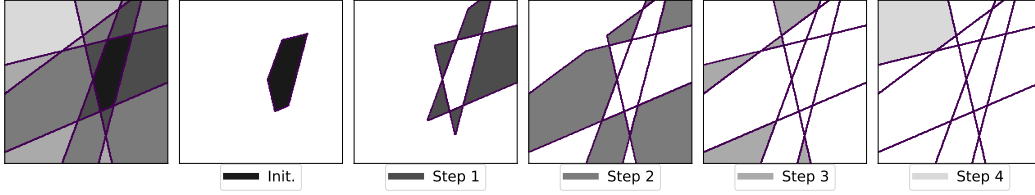

Figure 1: Recursive partition discovery for a DGN with $S = 2$ and $L = 2$, starting with an initial region obtained from a sampled latent vector $z$ (init). By walking on the faces of this region, neighboring regions sharing a common face are discovered (Step 1). Recursively repeating this process until no new region is discovered (Steps 2–4) provides the DGN latent space partition at left .

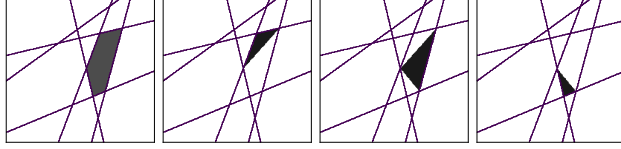

Figure 2: Triangulation $T(\omega)$ as per (9) of a polytopal region $\omega$ (left plot) obtained from the Delaunay Triangulation of the region vertices leading to 3 simplices (three right plots).

**Lemma 3.** *The operator $z \mapsto [q^1(z), \ldots, q^{L-1}(z)]$ is piecewise constant with a bijection between its image and $\Omega$.*

**Corollary 1.** *The polyhedral region $\omega$ is given by*

$$\omega = \bigcap_{\ell=1}^{L-1} \left\{ z \in \mathbb{R}^S : A_\omega^{1 \to \ell} z < -q^\ell(\omega) \odot b_\omega^{1 \to \ell} \right\},$$

*with $\odot$ the Hadamard product.*

The above result tells us that the pre-activation signs indicate which side of each hyperplane the region $\omega$ is located, which provides a direct way to compute the $\mathcal{H}$-representation of $\omega$ from $q(z)$ with $z \in \omega$. To obtain the entire partition $\Omega$, we propose a recursive scheme that starts from an initial region (or sample $z$) and walks on its faces to discover the neighboring regions. This process is repeated on the newly discovered regions until no new region is discovered. We detail this exploration procedure in Appendix A of the Supplemental Materials and illustrate it in Fig. 1.

**Gaussian Integration on $\omega$.** The Gaussian integral on a region $\omega$ (and its moments) cannot in general be obtained by direct integration unless $\omega$ is a rectangular region [46, 47] or is polytopal with at most $S$ faces [48]. In general, the DGN regions $\omega \in \Omega$ will have at least $S + 1$ faces, as they are closed polytopes in $\mathbb{R}^S$. To leverage the known integral forms, we propose to first decompose a DGN region $\omega$ into simplices ($S + 1$-face polytopes in our case [49]) and then further decompose each simplex into open polytopes with at most $S$ faces, which enables the use of [48]. In our case, we perform the simplex decomposition with the Delaunay triangulation [50] denoted as $T(\omega)$ with

$$T(\omega) \triangleq \{\Delta_1, \ldots, \Delta_{\text{Card}(T(\omega))}\}, \text{ with } \cup_{i=1}^{\text{Card}(T(\omega))} \Delta_i = \omega \text{ and } \Delta_i \cap \Delta_j = \emptyset, \forall i \neq j, \quad (9)$$

where each $\Delta_i$ is a simplex defined by the half-spaces $\Delta_i = \cap_{s=1}^{S+1} H_{i,j}$. This process is illustrated in Fig. 2. The decomposition of each simplex into open polytopes with less than $S + 1$ faces is performed by employing the standard inclusion-exclusion principle [51], leading to the following result.

**Lemma 4.** *The integral of any integrable function $g$ on a polytopal region $\omega \in \Omega$ can be decomposed into integration over open polytopes of at most $S$ faces via*

$$\int_\omega g(z)dz = \sum_{\Delta \in T(\omega)} \sum_{(s,V) \in H(\Delta)} s \int_V g(z)dz.$$

*with $H(\Delta_i) \triangleq \left\{ \left((-1)^{|J|+S}, \cap_{j \in J} H_{i,j}\right), J \subseteq \{1, \ldots, S+1\}, |J| \leq S \right\}$.*

From the above result, we can apply the known form of the Gaussian integral on a polytopal region with fewer than $S$ faces and obtain the form of the integral and moments as provided in Appendix B, where detailed pseudo code is also provided.

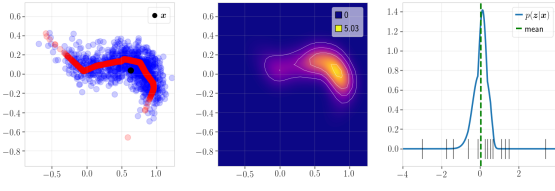

Figure 3: **Left:** Noiseless generated samples $g(z)$ in red and noisy samples $g(z) + \epsilon$ in blue, with $\Sigma_x = 0.1I$, $\Sigma_z = I$. **Middle:** marginal distribution $p(x)$ from (5). **Right:** the posterior distribution $p(z|x)$ from (6) (blue), its expectation (green) and the position of the region limits (black), with sample point $x$ depicted in black in the left figure.

**Computational Complexity:** Exact evaluation of the analytical DGN distributions is effected by (i) computing the partition, (ii) triangulating each partition region, and (iii) integrating on a region using Lemma 4. The first two steps have complexity growing with the latent space dimension and the number of regions. Even though their asymptotic complexity is linear with respect to the number of regions, one must recall that this latter quantity grows exponentially in the width and depth of a DN [19, 52, 53]. The third step of integrating as per Lemma 4 is computationally expensive, particularly with respect to the latent space dimension $S$. This is the current main practical limitation of performing the analytical computation of the DGN posterior (and thus the E-step). A more elaborated discussion plus several solutions are provided in the next section; see also Appendix I for the asymptotic computational complexity details.

**Visualization of the Marginal and Posterior Distributions.** To illustrate our theoretical development so far, we now visualize the posterior and marginal distributions of a randomly initialized DGN in a low-dimensional space $D = 2$ and with latent dimension $S = 1$. (See Appendix J for the architectural details of the DGN.) We depict the various distributions as well as the generated samples in Fig. 3. We also plot the posterior distribution based on one observation obtained via $g(z_0)$ given a sampled $z_0$ from the $z$ distribution and one noisy observation $g(z_0) + \epsilon_0$ given a noise realization $\epsilon_0$.

## 4 Expectation-Maximization Learning of Deep Generative Networks

We now derive an analytical Expectation-Maximization (EM) training algorithm for CPA DGNs based on the results of the previous sections. We then compare DGN training via EM and AVI and leverage the exact complete likelihood to perform model selection and study the VAE approximation error.

### 4.1 Expectation Step

The E-step infers the latent (unobserved) variables associated to the generation of each observation $x$ by taking the expectation of the log of the complete likelihood with respect to the posterior distribution (6). We denote the per-region moments of the DGN posterior (from Appendix B) by $\mathbb{E}_{z|x}[\mathbb{1}_{z \in \omega}] \triangleq e_\omega^0(x)$, $\mathbb{E}_{z|x}[z \mathbb{1}_{z \in \omega}] \triangleq e_\omega^1(x)$ and $\mathbb{E}_{z|x}[zz^T \mathbb{1}_{z \in \omega}] \triangleq E_\omega^2(x)$; we also have $e^1(x) \triangleq \mathbb{E}_{z|x}[z] = \sum_\omega e_\omega^1(x)$ and likewise for the second moment. We obtain the following E-step (the detailed derivations are in Appendix G.1):

$$
\begin{aligned}
E_{z|x}\left[\log\left(p(x|z)p(z)\right)\right] = &-\frac{1}{2}\log\left((2\pi)^{S+D}|\det(\Sigma_x)||\det(\Sigma_z)|\right) - \frac{1}{2}\mathrm{Tr}(\Sigma_z^{-1}E^2(x)) \\
&- \frac{1}{2}\left(x^T\Sigma_x^{-1}x - 2x^T\Sigma_x^{-1}\left(\sum_\omega A_\omega e_\omega^1(x) + b_\omega e_\omega^0(x)\right)\right. \\
&\left. + \sum_\omega\left[\mathrm{Tr}(A_\omega^T\Sigma_x^{-1}A_\omega E_\omega^2(x)) + (e_\omega^0 b_\omega + 2A_\omega e_\omega^1(x))^T\Sigma_x^{-1}b_\omega\right]\right).
\end{aligned}
$$

Note that the (per-region) moments involved in the E-step, such as $e_\omega^1(x)$, are taken with respect to the current parameters ($\theta = \{\Sigma_x, \Sigma_z, (W^\ell, v^\ell)_{\ell=1}^L\}$). That is, if gradient based optimization is leveraged to maximize the ELBO, then no gradient should be propagated through them. We can see from the above formula that the contributions of each region's affine parameters are weighted based on the posterior for each datum $x$. That is, for each input $x$, the posterior combines all of the per-region affine parameters as opposed to current forms of learning that leverage only the parameters involved in the specific region activated by the DGN input $z$.

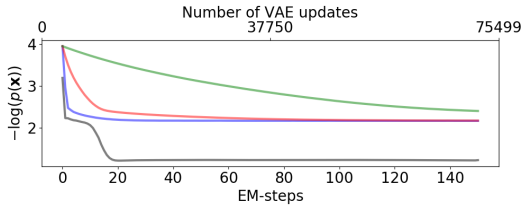

Figure 4: DGN training under EM (black) and VAE training with various learning rates for VAE (blue: 0.005, red: 0.001, green: 0.0001). In all cases, VAE converges to the maximum of its ELBO. The gap between the VAE and EM curves is due to the inability of the VAE's AVI to correctly estimate the true posterior, pushing the VAE's ELBO far from the true log-likelihood (recall (3)) and thus preventing it from precisely approximating the true data distribution.

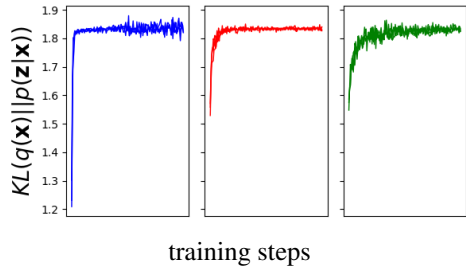

training steps

Figure 5: KL-divergence between a VAE variational distribution and the true DGN posterior when trained on a noisy circle dataset in $2D$ for 3 different learning rates. During learning, the DGN adapts such that $g(z) + \epsilon$ models the data distribution based on the VAE's estimated ELBO. As learning progresses, the true DGN posterior becomes harder to approximate by the VAE's variational distribution in the AVI process. As such, even in this toy dataset, the commonly employed Gaussian variational distribution is not rich enough to capture the multimodality of $p(z|x)$ from (6).

## 4.2 Maximization Step

Given the E-step, the ELBO can be maximized via some flavor of gradient based optimization. However, thanks to the analytical E-step and the Gaussian form of the involve distributions, there exists analytical form of this maximization process (M-step) leading to the analytical M-step for DGNs. The formulas for all of the DGN parameters are provided in Appendix G. We provide here the analytical form for the bias $v^{\ell*}$, for which we introduce $r_\omega^\ell(x)$ as the expected reconstruction error of the DGN as

$$r_\omega^\ell(x) \triangleq \left( x - \sum_{i \neq \ell} A_\omega^{i+1 \to L} D_\omega^i v^i \right) e_\omega^0(x) - A_\omega e_\omega^1(x) \quad \text{(expected residual without } v^\ell\text{)},$$

$$v^{\ell*} = \left( \sum_x \sum_\omega D_\omega^\ell A_\omega^{L \to \ell+1} \Sigma_x^{-1} A_\omega^{\ell+1 \to L} D_\omega^\ell \right)^{-1} \left( \sum_x \sum_{\omega \in \Omega} \underbrace{D_\omega^\ell A_\omega^{L \to \ell+1} \Sigma_x^{-1} r_\omega^\ell(x)}_{\text{residual back-propagated to layer } \ell} \right).$$

Some interesting observations can be made based on the analytical form of these updates. First, the bias update is based on the residual of the reconstruction error with a DGN whose bias has been removed; this residual is then backpropagated to the $\ell^{\text{th}}$ layer. The backpropagation is performed via the (transposed) backpropagation matrix as when performing gradient-based learning. Second, the updates of any parameter depend on each region parameter's contribution based on the posterior moments and integrals, similarly to any mixture model. Third, all of the updates are whitened based on the backpropagation (or forward propagation) correlation matrix $A_\omega^{\ell \to L}, \forall \omega, \forall \ell$. We study the impact of using a probabilistic prior on the layer weights such as Gaussian, Laplacian, or Uniform, which are related to the $\ell_2, \ell_1$ regularization and weight clipping techniques in Appendix H.

## 4.3 Empirical Validation and VAE Comparison

We now numerically validate the above EM-steps on a simple problem involving data points on a circle of radius 1 in 2D augmented with a Gaussian noise of standard deviation $0.05$. We depict the EM-training of a 2-layer DGN with width of $8$ against VAE training. In all cases the DGNs have the same architecture with same weight initialization; the dataset is also identical between models with the same noise realizations. Thanks to the analytical form of the marginals, we can compute the true ELBO (without variational estimation of the true posterior) for the VAE during its training to monitor its ability to fit the data distribution. We depict the evolution of the negative log-likelihood during training (EM-step for the EM training setting and VAE updates for VAEs) in Fig. 4.

We observe that EM training converges faster and to a lower negative log-likelihood. In addition, we see how all of the trained VAEs seem to converge to the same bound, which likely corresponds

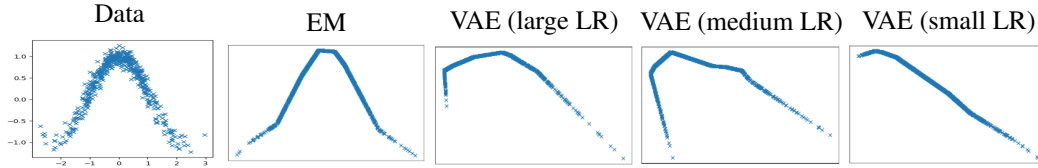

Figure 6: EM training of a DGN with latent dimension 1. We show only the generated continuous piecewise affine manifold $g(\boldsymbol{z})$ without the additional white noise $\epsilon$. We see how EM training of the DGN is able to fit the dataset, while VAE (with different learning rates (LR)) suffers from hyperparameter sensitivity and slow convergence. Training details and additional figures for this experiment are provided in Appendix J.

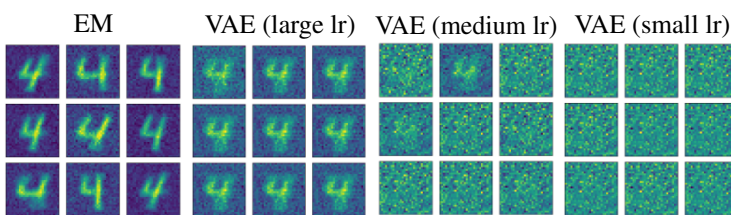

Figure 7: Reprise of Fig. 6 for MNIST data restricted to the digit 4, employing a 3-layer DGN with latent dimension of 1. Details of training and additional figures for this experiments are provided in Appendix J.

to the maximum of its ELBO, where the gap is induced by the use of a variational approximation of the true posterior. We confirm this by looking at the KL divergence between the true posterior and the AVI estimates of the VAE models during training in Fig. 5. We also experiment with another unidimensional manifold which is a localized subpart of a cosine function in $2D$ and a more complicated manifold that is MNIST constrained to the digit 4. We present the manifolds and the EM versus VAE learned manifolds in Fig. 6 and Fig. 7. We observe the ability of EM to fit the manifold while VAEs suffer from slow convergence and poor posterior approximation. Additional figures and experiments with various architectures are provided in Appendix J.

We thus observed that EM learning produces a much smaller negative log-likelihood and that providing better posterior estimates (improved AVI) is key to improve VAE performances. In particular, multimodal variational distributions should be considered for VAEs regardless of the data at hand. In fact, recall from (6) and Prop. 2 that the posterior is a mixture of truncated Gaussians with covariances based on $\boldsymbol{A}_\omega^T \boldsymbol{A}_\omega$.

## 5    Conclusions

We have derived the analytical form of the posterior, marginal, and conditional distributions for DGNs constructed using continuous piecewise affine nonlinearities with Gaussian output and latent distributions. This has enabled us to derive the EM-learning algorithm for DGNs that not only converges faster than state-of-the-art VAE training but also to a higher likelihood. Our proposed methodology also applies to more general distributions, requiring them only to be conjugate priors in order to obtain an analytical solution. Our analytical forms can be leveraged to improve the variational distribution of VAEs, understand the form of analytical weight updates, study how a DGN infers the latent variable $\boldsymbol{z}$ from $\boldsymbol{x}$, and leverage standard statistical tools to perform model selection, anomaly detection and beyond.

### Broader Impacts

We have derived the analytical form of the posterior, marginal and conditional distributions of DGNs based on CPA architectures. Our approach provides an approximation-free alternative to VAEs to train DGNs. In addition to improving DGN algorithms, our analytical forms will enable researchers to probe more deeply into the inner workings of DGNs and VAE, making them more interpretable and thus trustworthy. Our calculations will also enable accurate anomaly detection and model selection, which should find wide application in sensitive applications where accurately computing the probability of a data point is crucial.

### Acknowledgments

RB and RB were supported by NSF grants CCF-1911094, IIS-1838177, and IIS-1730574; ONR grants N00014-18-12571 and N00014-17-1-2551; AFOSR grant FA9550-18-1-0478; DARPA grant G001534-7500; and a Vannevar Bush Faculty Fellowship, ONR grant N00014-18-1-2047.

## Footnotes

[1]Note that we do not require that $S < D$.

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
