[Supplementary Material]

# Supplementary Material

## A    Computing the Latent Space Partition

In this section we first introduce notations and demonstrate how to express a region $\omega$ of the partition $\Omega$ as a polytope defined by a system of inequalities, and then leverage this formulation to demonstrate how to obtain $\Omega$ by recursively exploring neighboring regions starting from a random point/region.

**Regions as Polytopes** To represent the regions $\omega \in \Omega$ as a polytope via a system of inequalities we need to recall from (1) that the input-output mapping is defined on each region by the affine parameters $A_\omega, B_\omega$ themselves obtained by composition of MASOs. Each layer pre-activation (feature map prior application of the nonlinearity) is denoted by $\boldsymbol{h}^\ell(\boldsymbol{z}) \in \mathbb{R}^{D^\ell}, \ell = 1, \dots, L-1$ and given by $\boldsymbol{h}^\ell(\boldsymbol{x}) = \boldsymbol{A}_\omega^{1 \to \ell} \boldsymbol{z} + \boldsymbol{b}_\omega^{1 \to \ell}$, with up-to-layer $\ell$ affine parameters

$$\boldsymbol{A}_\omega^{1\to\ell} \triangleq \boldsymbol{W}^\ell \boldsymbol{D}_\omega^{\ell-1} \boldsymbol{W}^{\ell-1} \dots \boldsymbol{D}_\omega^1 \boldsymbol{W}^1, \qquad \boldsymbol{A}_\omega^{1\to\ell} \in \mathbb{R}^{D^\ell \times S}, \tag{10}$$

$$\boldsymbol{b}_\omega^{1\to\ell-1} \triangleq \boldsymbol{v}^\ell + \sum_{i=1}^\ell \boldsymbol{W}^\ell \boldsymbol{D}_\omega^{\ell-1} \boldsymbol{W}^{\ell-1} \dots \boldsymbol{D}_\omega^i \boldsymbol{v}^i, \qquad \boldsymbol{b}_\omega^{1\to\ell} \in \mathbb{R}^{D^\ell}, \tag{11}$$

which depend on the region $\omega$ in the latent space [2]. Notice that we have in particular $\boldsymbol{A}_\omega^L = \boldsymbol{A}_\omega$ and $\boldsymbol{b}_\omega^L = \boldsymbol{b}_\omega$, the entire DGN affine parameters from (2) on region $\omega$. The regions depend on the signs of the pre-activations defined as $\boldsymbol{q}^\ell(\boldsymbol{z}) = \text{sign}(\boldsymbol{h}^\ell(\boldsymbol{z}))$ due to the used activation function behaving linearly as long as the feature maps preserve the same sign. This holds for (leaky-)ReLU or absolute value, for max-pooling we would need to look at the argmax position of each pooling window, as pooling is rare in DGN we focus here on DN without max-pooling; let $\boldsymbol{q}^{\text{all}}(\boldsymbol{z}) \triangleq [(\boldsymbol{q}^{L-1}(\boldsymbol{z}))^T, \dots, (\boldsymbol{q}^1(\boldsymbol{z}))^T]^T$ collect all the per layer sign operators without the last layer as it does not apply any activation.

**Lemma 5.**  *The $\boldsymbol{q}^{\text{all}}$ operator is piecewise constant and there is a bijection between $\Omega$ and $Im(\boldsymbol{q})$.*

The above demonstrates the equivalence of knowing $\omega$ in which an input $\boldsymbol{z}$ belongs to and knowing the sign pattern of the feature maps associated to $\boldsymbol{z}$; we will thus use interchangeably $\boldsymbol{q}^{\text{all}}(\boldsymbol{z}), \boldsymbol{z} \in \omega$ and $\boldsymbol{q}^{\text{all}}(\omega)$. From this, we see that the pre-activation signs and the regions are tied together. We can now leverage this result and provide the explicit region $\omega$ as a polytope via its sytem of inequality, to do so we need to collect the per-layer slopes and biases into

$$\boldsymbol{A}_\omega^{\text{all}} = \begin{bmatrix} \boldsymbol{A}_\omega^{1\to L-1} \\ \dots \\ \boldsymbol{A}_\omega^{1\to 1} \end{bmatrix}, \ \boldsymbol{b}_\omega^{\text{all}} = \begin{bmatrix} \boldsymbol{b}_\omega^{1\to L-1} \\ \dots \\ \boldsymbol{b}_\omega^{1\to 1} \end{bmatrix}, \ \boldsymbol{A}_\omega^{\text{all}} \in \mathbb{R}^{(\prod_{\ell=1}^{L-1} D^\ell) \times S}, \boldsymbol{b}_\omega^{\text{all}} \in \mathbb{R}^{\prod_{\ell=1}^{L-1} D^\ell}. \tag{12}$$

**Corollary 2.** *The $\mathcal{H}$-representation of the polyhedral region $\omega$ is given by*

$$\omega = \{\boldsymbol{z} \in \mathbb{R}^S : \boldsymbol{A}_\omega^{\text{all}} \boldsymbol{z} < -\boldsymbol{q}^{\text{all}}(\omega) \odot \boldsymbol{b}_\omega^{\text{all}}\} = \bigcap_{\ell=1}^{L-1} \{\boldsymbol{z} \in \mathbb{R}^S : \boldsymbol{A}_\omega^{1\to\ell} \boldsymbol{z} < -\boldsymbol{q}^\ell(\omega) \odot \boldsymbol{b}_\omega^{1\to\ell}\}, \tag{13}$$

*with $\odot$ the Hadamard product.*

From the above, it is clear that the sign locates in which side of each hyperplane the region is located. We now have a direct way to obtain the polytope $\omega$ from its sign pattern $\boldsymbol{q}^{\text{all}}(\omega)$ or equivalently from an input $\boldsymbol{z} \in \omega$; the only task left is to obtain the entire partition $\Omega$ collecting all the DN regions, which we now propose to do via a simple scheme.

**Partition Cells Enumeration.** The search for all cells in a partition is known as the cell enumeration problem and has been extensively studied in the context of speicific partitions such as hypreplane arrangements [54–56]. In our case however, the set of inequalitites of different regions changes. In fact, for any neighbour region, not only the sign pattern $\boldsymbol{q}^{\text{all}}$ will change but also $\boldsymbol{A}_\omega^{\text{all}}$ and $\boldsymbol{b}_\omega^{\text{all}}$ due to the

composition of layers. In fact, changing one activation state say $-1$ to $1$ for a specific unit at layer $\ell$ will alter the affine parameters from (10) and (11) due to the layer composition. As such, we propose to enumerate all the cells $\omega \in \Omega$ with a deterministic algorithm that starts from an intial region and recursively explores its neighbouring cells untill all have been visited while recomputing the inequality system at each step. To do so, consider the initial region $\omega_0$. First, one finds all the non-redundant inequalities of the inequality system (13), the remaining inequalities define the faces of the polytope $\omega$. Second, one obtains any of the neighbouring regions sharing a face with $\omega_0$ by switching the sign in the entry of $\boldsymbol{q}(\omega_0)$ corresponding to the considered face. Repeat this for all non-redundant inequalities to obtain all the adjacent regions to $\omega_0$ sharing a face with it. Each altered code defines an adjacent region and its sytem of inequality can be obtain as per Lemma 2. Doing so for all the faces of the initial region and then iterating this process on all the newly discovered regions will enumerate the entire partition $\Omega$. We summarize this in Algo 1 in the appendix and illustrate this recursive procedure in Fig. 1.

We now have each cell as a polytope and enumerated the partition $\Omega$, we can now turn into the computation of the marginal and posterior DGN distributions.

## B   Analytical Moments for truncated Gaussian

To lighten the derivation, we introduce extend the $[.]$ indexing operator such that for example for a matrix, $[.]_{-k,.}$ means that all the rows but the $k^{\text{th}}$ are taken, and all columns are taken. Also, $[.]_{(k,l),.}$ means that only the $k^{\text{th}}$ and $l^{\text{th}}$ rows are taken and all the columns. Let also introduce the following quantities

$$[F(\boldsymbol{a}, \boldsymbol{\Sigma})]_k = \phi\left([\boldsymbol{a}]_k; 0, [\boldsymbol{\Sigma}]_{k,k}\right) \Phi_{[[\boldsymbol{a}]_{-k}, \infty)}\left(\boldsymbol{\mu}(k), \boldsymbol{\Sigma}(k)\right)$$

$$[G(\boldsymbol{a}, \boldsymbol{\Sigma})]_{k,l} = \phi\left([\boldsymbol{a}]_{(k,l)}; 0, [\boldsymbol{\Sigma}]_{(k,j),(k,j)}\right) \Phi_{[[\boldsymbol{a}]_{-(k,l)}, \infty)}\left(\boldsymbol{\mu}((k,l)), \boldsymbol{\Sigma}((k,l))\right)$$

$$H(\boldsymbol{a}, \boldsymbol{\Sigma}) = G(\boldsymbol{a}, \Sigma) + \text{diag}\left(\frac{\boldsymbol{l} \odot F(\boldsymbol{l}, \boldsymbol{\Sigma}) - (\boldsymbol{\Sigma} \odot G(\boldsymbol{l}, \Sigma))\mathbf{1}}{\text{diag}(\boldsymbol{\Sigma})}\right)$$

with $\boldsymbol{\mu}(u) = [\boldsymbol{\Sigma}]_{-u,u}[\boldsymbol{\Sigma}]_{u,u}^{-1}[\boldsymbol{a}]_u$, and $\boldsymbol{\Sigma}(u) = [\boldsymbol{\Sigma}]_{-u,-u} - [\boldsymbol{\Sigma}]_{-u,u}[\boldsymbol{\Sigma}]_{u,u}^{-1}[\boldsymbol{\Sigma}]_{-u,u}^T$. Thanks to the above form, we can now obtain the integral $e_\omega^0(\boldsymbol{\Sigma}) \triangleq \Phi_\omega(\boldsymbol{0}, \boldsymbol{\Sigma})$ and the first two moments of a centered truncated gaussian $e_\omega^1(\boldsymbol{\Sigma}) \triangleq \int_\omega \boldsymbol{z}\phi(\boldsymbol{z}; \boldsymbol{0}, \boldsymbol{\Sigma})$ and $\boldsymbol{E}_\omega^2(\boldsymbol{\Sigma}) \triangleq \int_\omega \boldsymbol{z}\boldsymbol{z}^T\phi(\boldsymbol{z}; \boldsymbol{0}, \boldsymbol{\Sigma})$

**Corollary 3.** *The integral and first two moments of a centered truncated gaussian are given by*

$$e_\omega^0(\boldsymbol{\Sigma}) = \sum_{\Delta \in T(\omega)} \sum_{(s,C) \in T(\Delta)} s\Phi_{[\boldsymbol{l}(C), \infty)}\left(0, R_c \boldsymbol{\Sigma} R_c^T\right) d\boldsymbol{z}, \tag{14}$$

$$e_\omega^1(\boldsymbol{\Sigma}) = \boldsymbol{\Sigma} \sum_{\Delta \in T(\omega)} \sum_{(s,C) \in T(\Delta)} sR_C^T F(\boldsymbol{l}_{\omega,c}, R_c \boldsymbol{\Sigma} R_c^T), \tag{15}$$

$$\boldsymbol{E}_\omega^2(\boldsymbol{\Sigma}) = \boldsymbol{\Sigma}\left(\sum_{\Delta \in T(\omega)} \sum_{(s,C) \in T(\Delta)} sR_C^T(H(\boldsymbol{l}_{\omega,C}, R_c \boldsymbol{\Sigma} R_c^T))R_C\right)\boldsymbol{\Sigma} + e_\omega^0(\boldsymbol{\Sigma})\boldsymbol{\Sigma} \tag{16}$$

To simplify notations let consider the following notation of the posterior (6) where are incorporate the terms independent of $\boldsymbol{z}$ into

$$\alpha_\omega(\boldsymbol{x}) = \frac{\phi(\boldsymbol{x}; B_\omega, \boldsymbol{\Sigma_x} + A_\omega \boldsymbol{\Sigma_z} A_\omega^T)}{\sum_\omega \phi(\boldsymbol{x}; B_\omega, \boldsymbol{\Sigma_x} + A_\omega \boldsymbol{\Sigma_z} A_\omega^T)\Phi_\omega(\boldsymbol{\mu}_\omega(\boldsymbol{x}), \boldsymbol{\Sigma}_\omega)}, \tag{17}$$

leading to $p(\boldsymbol{z}|\boldsymbol{x}) = \sum_{\omega \in \Omega} \delta_\omega(\boldsymbol{z})\alpha_\omega(\boldsymbol{x})\phi(\boldsymbol{z}; \boldsymbol{\mu}_\omega(\boldsymbol{x}), \boldsymbol{\Sigma}_\omega)$.

**Theorem 2.** *The first (per region) moments of the DGN posterior are given by*

$$\mathbb{E}_{\boldsymbol{z}|\boldsymbol{x}}[\mathbb{1}_{\boldsymbol{z} \in \omega}] = \alpha_\omega(\boldsymbol{x})e_\omega^0(\boldsymbol{\Sigma}_\omega),$$

$$\mathbb{E}_{\boldsymbol{z}|\boldsymbol{x}}[\boldsymbol{z}\mathbb{1}_{\boldsymbol{z} \in \omega}] = \alpha_\omega(\boldsymbol{x})\left(e_{\omega - \boldsymbol{\mu}_\omega(\boldsymbol{x})}^1(\boldsymbol{\Sigma}_\omega) + e_{\omega - \boldsymbol{\mu}_\omega(\boldsymbol{x})}^0(\boldsymbol{\Sigma}_\omega)\boldsymbol{\mu}_\omega(\boldsymbol{x})\right)$$

$$\mathbb{E}_{\boldsymbol{z}|\boldsymbol{x}}[\boldsymbol{z}\boldsymbol{z}^T \mathbb{1}_{\boldsymbol{z}\in\omega}] = \alpha_\omega(\boldsymbol{x})\big(\boldsymbol{E}^2_{\omega-\boldsymbol{\mu}_\omega(\boldsymbol{x})}(\boldsymbol{\Sigma}_\omega) + \boldsymbol{e}^1_{\omega-\boldsymbol{\mu}_\omega(\boldsymbol{x})}(\boldsymbol{\Sigma}_\omega)\boldsymbol{\mu}_\omega(\boldsymbol{x})^T$$
$$+ \boldsymbol{\mu}_{\omega-\boldsymbol{\mu}_\omega(\boldsymbol{x})}(\boldsymbol{x})\boldsymbol{e}^1_{\omega-\boldsymbol{\mu}_\omega(\boldsymbol{x})}(\boldsymbol{x})^T + \boldsymbol{\mu}_\omega(\boldsymbol{x})\boldsymbol{\mu}_\omega(\boldsymbol{x})^T e^0_\omega(\boldsymbol{x})\big)$$

*which we denote* $\mathbb{E}_{\boldsymbol{z}|\boldsymbol{x}}[\mathbb{1}_{\boldsymbol{z}\in\omega}] \triangleq e^0_\omega(\boldsymbol{x})$, $\mathbb{E}_{\boldsymbol{z}|\boldsymbol{x}}[\boldsymbol{z}\mathbb{1}_{\boldsymbol{z}\in\omega}] \triangleq \boldsymbol{e}^1_\omega(\boldsymbol{x})$ *and* $\mathbb{E}_{\boldsymbol{z}|\boldsymbol{x}}[\boldsymbol{z}\boldsymbol{z}^T\mathbb{1}_{\boldsymbol{z}\in\omega}] \triangleq \boldsymbol{E}^2_\omega(\boldsymbol{x})$. *(Proof in F.9.)*

## C   Implementation Details

The Delaunay triangulation needs the $\mathcal{V}$-representation of $\omega$, the vertices which convex hull form the region [57]. Given that we have the $\mathcal{H}$-representation, finding the vertices is known as the vertex enumeration problem [58]. To compute the triangulation we use the Python scipy [59] implementation which interfaces the C/C++ Qhull implementation [60]. To compute the $\mathcal{H} \mapsto \mathcal{V}$ representation and vice-versa we leverage pycddlib [3] which interfaces the C/C++ cddlib library [4] employing the double description method [61].

## D   Figures

We demonstrate here additional figures for the posterior and marginal distribution of a DGN.

Figure 8: Additional random DGNs with their samples, the posterior and the marginal distributions.

# E  Algorithms

**Algorithm 1:** SearchRegion

---

**Data:** Starting region $\omega$ and $\boldsymbol{q}(\omega)$, initial set $(\Omega)$
**Result:** Updated $\Omega$
**if** $\omega \notin \Omega$ **then**
|     $\Omega \leftarrow \Omega \cup \{\omega\}$;
**else**
|     Quit
**end**
$I = reduce(A_\omega^{\text{all}}, B_\omega^{\text{all}})$;
**for** $i \in I$ **do**
|     SearchRegion(flip($\boldsymbol{q}(\omega), i$), $\Omega$);
**end**

---

# F  Proofs

In this section we provide all the proofs for the main paper theoretical claims. In particular we will go through the derivations of the per region posterior first moments and then the derivation of the expectation and maximization steps.

## F.1  Proof of Lemma 1

*Proof.* The proof consists of expressing the conditional distribution and using the properties of DGN with piecewise affine nonlinearities. We are able to split the distribution into a mixture model as follows:

$$
\begin{aligned}
p(\boldsymbol{x}|\boldsymbol{z}) &= \frac{1}{(2\pi)^{D/2}\sqrt{|\det \Sigma_{\boldsymbol{x}}|}} e^{-\frac{1}{2}(\boldsymbol{x}-g(\boldsymbol{z}))^T \Sigma_{\boldsymbol{x}}^{-1}(\boldsymbol{x}-g(\boldsymbol{z}))} \\
&= \frac{1}{(2\pi)^{D/2}\sqrt{|\det \Sigma_{\boldsymbol{x}}|}} e^{-\frac{1}{2}(\boldsymbol{x}-\sum_{\omega\in\Omega} \mathbb{1}_{\boldsymbol{z}\in\omega}(A_\omega \boldsymbol{z}+B_\omega))^T \Sigma_{\boldsymbol{x}}^{-1}(\boldsymbol{x}-\sum_{\omega\in\Omega} \mathbb{1}_{\boldsymbol{z}\in\omega}(A_\omega \boldsymbol{z}+B_\omega))} \\
&= \frac{1}{(2\pi)^{D/2}\sqrt{|\det \Sigma_{\boldsymbol{x}}|}} e^{-\frac{1}{2}\sum_{\omega\in\Omega} \mathbb{1}_{\boldsymbol{z}\in\omega}(\boldsymbol{x}-(A_\omega \boldsymbol{z}+B_\omega))^T \Sigma_{\boldsymbol{x}}^{-1}(\boldsymbol{x}-(A_\omega \boldsymbol{z}+B_\omega))} \\
&= \sum_{\omega\in\Omega} \mathbb{1}_{\boldsymbol{z}\in\omega} \frac{1}{(2\pi)^{D/2}\sqrt{|\det \Sigma_{\boldsymbol{x}}|}} e^{-\frac{1}{2}(\boldsymbol{x}-(A_\omega \boldsymbol{z}+B_\omega))^T \Sigma_{\boldsymbol{x}}^{-1}(\boldsymbol{x}-(A_\omega \boldsymbol{z}+B_\omega))} \\
&= \sum_{\omega\in\Omega} \mathbb{1}_{\boldsymbol{z}\in\omega} \phi(\boldsymbol{x}|A_\omega \boldsymbol{z}+B_\omega, \Sigma_{\boldsymbol{x}})
\end{aligned}
$$

$\square$

## F.2  Proof of Proposition 1

*Proof.* This result is direct by noticing that the probability to obtain a specific region slope and bias is the probability that the sampled latent vector lies in the corresponding region. This probability is obtained simply by integrating the latent gaussian distribution on the region. We obtain the result of the proposition. $\square$

## F.3  Proof of Theorem 1

*Proof.* For the first part, we simply leverage the known result from linear Gaussian models [62] stating that

$$
p(\boldsymbol{z}|\boldsymbol{x}) = \frac{p(\boldsymbol{x}|\boldsymbol{z})p(\boldsymbol{z})}{p(\boldsymbol{x})}
$$

$$
= \frac{1}{p(\boldsymbol{x})} \frac{e^{-\frac{1}{2}(\boldsymbol{x}-g(\boldsymbol{z}))^T \Sigma_{\boldsymbol{x}}^{-1}(\boldsymbol{x}-g(\boldsymbol{z}))}}{(2\pi)^{D/2}\sqrt{|\det(\Sigma_{\boldsymbol{x}})|}} \frac{e^{-\frac{1}{2}(\boldsymbol{z}-\boldsymbol{\mu})^T \Sigma_{\boldsymbol{z}}^{-1}(\boldsymbol{z}-\boldsymbol{\mu})}}{(2\pi)^{S/2}\sqrt{|\det(\Sigma_{\boldsymbol{z}})|}}
$$

$$
= \frac{1}{p(\boldsymbol{x})} \left( \sum_{\omega \in \Omega} \mathbb{1}_{\boldsymbol{z} \in \omega} \frac{e^{-\frac{1}{2}(\boldsymbol{x}-A_\omega \boldsymbol{z}-B_\omega)^T \Sigma_{\boldsymbol{x}}^{-1}(\boldsymbol{x}-A_\omega \boldsymbol{z}-B_\omega)}}{(2\pi)^{D/2}\sqrt{|\det(\Sigma_{\boldsymbol{x}})|}} \right) \frac{e^{-\frac{1}{2}(\boldsymbol{z}-\boldsymbol{\mu})^T \Sigma_{\boldsymbol{z}}^{-1}(\boldsymbol{z}-\boldsymbol{\mu})}}{(2\pi)^{S/2}\sqrt{|\det(\Sigma_{\boldsymbol{z}})|}}
$$

$$
= \frac{1}{p(\boldsymbol{x})} \sum_{\omega \in \Omega} \mathbb{1}_{\boldsymbol{z} \in \omega} \frac{e^{-\frac{1}{2}(\boldsymbol{x}-A_\omega \boldsymbol{z}-B_\omega)^T \Sigma_{\boldsymbol{x}}^{-1}(\boldsymbol{x}-A_\omega \boldsymbol{z}-B_\omega)-\frac{1}{2}\boldsymbol{z}^T \Sigma_{\boldsymbol{z}}^{-1} \boldsymbol{z}}}{(2\pi)^{(S+D)/2}\sqrt{|\det(\Sigma_{\boldsymbol{x}})||\det(\Sigma_{\boldsymbol{z}})|}}
$$

$$
= \frac{1}{p(\boldsymbol{x})} \sum_{\omega \in \Omega} \mathbb{1}_{\boldsymbol{z} \in \omega} \frac{e^{-\frac{1}{2}((\boldsymbol{x}-B_\omega)-A_\omega \boldsymbol{z})^T \Sigma_{\boldsymbol{x}}^{-1}((\boldsymbol{x}-B_\omega)-A_\omega \boldsymbol{z})-\frac{1}{2}\boldsymbol{z}^T \Sigma_{\boldsymbol{z}}^{-1} \boldsymbol{z}}}{(2\pi)^{(S+D)/2}\sqrt{|\det(\Sigma_{\boldsymbol{x}})||\det(\Sigma_{\boldsymbol{z}})|}}
$$

$$
= \frac{1}{p(\boldsymbol{x})} \sum_{\omega \in \Omega} \mathbb{1}_{\boldsymbol{z} \in \omega} \frac{e^{-\frac{1}{2}((A_\omega^T \Sigma_{\boldsymbol{x}}^{-1} A_\omega+\Sigma_{\boldsymbol{z}}^{-1})^{-1} A_\omega^T \Sigma_{\boldsymbol{x}}^{-1}(\boldsymbol{x}-B_\omega)-\boldsymbol{z})^T (A_\omega^T \Sigma_{\boldsymbol{x}}^{-1} A_\omega+\Sigma_{\boldsymbol{z}}^{-1})((A_\omega^T \Sigma_{\boldsymbol{x}}^{-1} A_\omega+\Sigma_{\boldsymbol{z}}^{-1})^{-1} A_\omega^T \Sigma_{\boldsymbol{x}}^{-1}(\boldsymbol{x}-B_\omega)-\boldsymbol{z})}}{(2\pi)^{(S+D)/2}\sqrt{|\det(\Sigma_{\boldsymbol{x}})||\det(\Sigma_{\boldsymbol{z}})|}}
$$
$$
\times\, e^{-\frac{1}{2}((\boldsymbol{x}-B_\omega)^T \Sigma_{\boldsymbol{x}}^{-1}(\boldsymbol{x}-B_\omega))+\frac{1}{2}((\boldsymbol{x}-B_\omega)^T \Sigma_{\boldsymbol{x}}^{-1} A_\omega (A_\omega^T \Sigma_{\boldsymbol{x}}^{-1} A_\omega+\Sigma_{\boldsymbol{z}}^{-1})^{-1} A_\omega^T \Sigma_{\boldsymbol{x}}^{-1}(\boldsymbol{x}-B_\omega))}
$$

$$
= \frac{1}{p(\boldsymbol{x})} \sum_{\omega \in \Omega} \mathbb{1}_{\boldsymbol{z} \in \omega} \frac{e^{-\frac{1}{2}((A_\omega^T \Sigma_{\boldsymbol{x}}^{-1} A_\omega+\Sigma_{\boldsymbol{z}}^{-1})^{-1} A_\omega^T \Sigma_{\boldsymbol{x}}^{-1}(\boldsymbol{x}-B_\omega)-\boldsymbol{z})^T (A_\omega^T \Sigma_{\boldsymbol{x}}^{-1} A_\omega+\Sigma_{\boldsymbol{z}}^{-1})((A_\omega^T \Sigma_{\boldsymbol{x}}^{-1} A_\omega+\Sigma_{\boldsymbol{z}}^{-1})^{-1} A_\omega^T \Sigma_{\boldsymbol{x}}^{-1}(\boldsymbol{x}-B_\omega)-\boldsymbol{z})}}{(2\pi)^{(S+D)/2}\sqrt{|\det(\Sigma_{\boldsymbol{x}})||\det(\Sigma_{\boldsymbol{z}})|}}
$$
$$
\times\, e^{-\frac{1}{2}((\boldsymbol{x}-B_\omega)^T (\Sigma_{\boldsymbol{x}}^{-1}-\Sigma_{\boldsymbol{x}}^{-1} A_\omega (A_\omega^T \Sigma_{\boldsymbol{x}}^{-1} A_\omega+\Sigma_{\boldsymbol{z}}^{-1})^{-1} A_\omega^T \Sigma_{\boldsymbol{x}}^{-1})(\boldsymbol{x}-B_\omega))}
$$

$$
= \frac{1}{p(\boldsymbol{x})} \sum_{\omega \in \Omega} \mathbb{1}_{\boldsymbol{z} \in \omega} \frac{e^{-\frac{1}{2}((A_\omega^T \Sigma_{\boldsymbol{x}}^{-1} A_\omega+\Sigma_{\boldsymbol{z}}^{-1})^{-1} A_\omega^T \Sigma_{\boldsymbol{x}}^{-1}(\boldsymbol{x}-B_\omega)-\boldsymbol{z})^T (A_\omega^T \Sigma_{\boldsymbol{x}}^{-1} A_\omega+\Sigma_{\boldsymbol{z}}^{-1})((A_\omega^T \Sigma_{\boldsymbol{x}}^{-1} A_\omega+\Sigma_{\boldsymbol{z}}^{-1})^{-1} A_\omega^T \Sigma_{\boldsymbol{x}}^{-1}(\boldsymbol{x}-B_\omega)-\boldsymbol{z})}}{(2\pi)^{(S+D)/2}\sqrt{|\det(\Sigma_{\boldsymbol{x}})||\det(\Sigma_{\boldsymbol{z}})|}}
$$
$$
\times\, e^{-\frac{1}{2}((\boldsymbol{x}-B_\omega)^T (\Sigma_{\boldsymbol{x}}+A_\omega \Sigma_{\boldsymbol{z}} A_\omega^T)^{-1}(\boldsymbol{x}-B_\omega))}
$$

$$
= \frac{1}{p(\boldsymbol{x})} \sum_{\omega \in \Omega} \mathbb{1}_{\boldsymbol{z} \in \omega} \frac{e^{-\frac{1}{2}(\boldsymbol{\mu}_\omega(\boldsymbol{x})-\boldsymbol{z})^T \Sigma_\omega^{-1}(\boldsymbol{\mu}_\omega(\boldsymbol{x})-\boldsymbol{z})}}{(2\pi)^{(S+D)/2}\sqrt{|\det(\Sigma_{\boldsymbol{x}})||\det(\Sigma_{\boldsymbol{z}})|}} e^{-\frac{1}{2}((\boldsymbol{x}-B_\omega)^T (\Sigma_{\boldsymbol{x}}+A_\omega \Sigma_{\boldsymbol{z}} A_\omega^T)^{-1}(\boldsymbol{x}-B_\omega))}
$$

with $\boldsymbol{\mu}_\omega(\boldsymbol{x}) = \Sigma_\omega A_\omega^T \Sigma_{\boldsymbol{x}}^{-1}(\boldsymbol{x}-B_\omega)$ and $\Sigma_\omega = (A_\omega^T \Sigma_{\boldsymbol{x}}^{-1} A_\omega + \Sigma_{\boldsymbol{z}}^{-1})^{-1}$ as a result it corresponds to a mixture of truncated gaussian, each living on $\omega$. Now we determine the renormalization constant:

$$
p(\boldsymbol{x}) = \int p(\boldsymbol{x}|\boldsymbol{z})p(\boldsymbol{z})d\boldsymbol{z}
$$

$$
= \sum_{\omega \in \Omega} \int_\omega \mathbb{1}_{\boldsymbol{z} \in \omega} \frac{e^{-\frac{1}{2}(\boldsymbol{\mu}_\omega(\boldsymbol{x})-\boldsymbol{z})^T \Sigma_\omega^{-1}(\boldsymbol{\mu}_\omega(\boldsymbol{x})-\boldsymbol{z})}}{(2\pi)^{(S+D)/2}\sqrt{|\det(\Sigma_{\boldsymbol{x}})||\det(\Sigma_{\boldsymbol{z}})|}} e^{-\frac{1}{2}((\boldsymbol{x}-B_\omega)^T (\Sigma_{\boldsymbol{x}}+A_\omega \Sigma_{\boldsymbol{z}} A_\omega^T)^{-1}(\boldsymbol{x}-B_\omega))} d\boldsymbol{z}
$$

$$
= \sum_{\omega \in \Omega} \mathbb{1}_{\boldsymbol{z} \in \omega} \frac{e^{-\frac{1}{2}((\boldsymbol{x}-B_\omega)^T (\Sigma_{\boldsymbol{x}}+A_\omega \Sigma_{\boldsymbol{z}} A_\omega^T)^{-1}(\boldsymbol{x}-B_\omega))}}{(2\pi)^{D/2}\sqrt{|\det(\Sigma_{\boldsymbol{x}})||\det(\Sigma_{\boldsymbol{z}})|}} \sqrt{\det(\Sigma_\omega)} \int_\omega \phi(\boldsymbol{z};\boldsymbol{\mu}_\omega(\boldsymbol{x}),\Sigma_\omega)d\boldsymbol{z}
$$

$$
= \sum_{\omega \in \Omega} \mathbb{1}_{\boldsymbol{z} \in \omega} \frac{e^{-\frac{1}{2}((\boldsymbol{x}-B_\omega)^T (\Sigma_{\boldsymbol{x}}+A_\omega \Sigma_{\boldsymbol{z}} A_\omega^T)^{-1}(\boldsymbol{x}-B_\omega))}}{(2\pi)^{D/2}\sqrt{|\det(\Sigma_{\boldsymbol{x}})||\det(\Sigma_{\boldsymbol{z}})|}} \sqrt{\det(\Sigma_\omega)} \Phi_\omega(\boldsymbol{\mu}_\omega(\boldsymbol{x}),\Sigma_\omega)
$$

$$
= \sum_{\omega \in \Omega} \mathbb{1}_{\boldsymbol{z} \in \omega} \frac{\sqrt{\det(\Sigma_{\boldsymbol{x}}+A_\omega \Sigma_{\boldsymbol{z}} A_\omega^T)\det(\Sigma_\omega)}}{\sqrt{|\det(\Sigma_{\boldsymbol{x}})||\det(\Sigma_{\boldsymbol{z}})|}} \phi(\boldsymbol{x};B_\omega,\Sigma_{\boldsymbol{x}}+A_\omega \Sigma_{\boldsymbol{z}} A_\omega^T)\Phi_\omega(\boldsymbol{\mu}_\omega(\boldsymbol{x}),\Sigma_\omega),
$$

now using the Matrix determinant lemma [63] we have that $\det(\Sigma_{\boldsymbol{x}}+A_\omega \Sigma_{\boldsymbol{z}} A_\omega^T) = \det(\Sigma_{\boldsymbol{z}}^{-1}+A_\omega^T \Sigma_{\boldsymbol{x}}^{-1} A_\omega)\det(\Sigma_{\boldsymbol{x}})\det(\Sigma_{\boldsymbol{z}})$ leading to

$$
p(\boldsymbol{x}) = \sum_\omega \phi(\boldsymbol{x};B_\omega,\Sigma_{\boldsymbol{x}}+A_\omega \Sigma_{\boldsymbol{z}} A_\omega^T)\Phi_\omega(\boldsymbol{\mu}_\omega(\boldsymbol{x}),\Sigma_\omega),
$$

$$p(\boldsymbol{z}|\boldsymbol{x}) = \sum_{\omega} \delta_{\omega}(\boldsymbol{z}) \frac{\phi(\boldsymbol{x}; B_{\omega}, \Sigma_{\boldsymbol{x}} + A_{\omega}\Sigma_{\boldsymbol{z}}A_{\omega}^T)\phi(\boldsymbol{z}; \boldsymbol{\mu}_{\omega}(\boldsymbol{x}), \Sigma_{\omega})}{\sum_{\omega} \phi(\boldsymbol{x}; B_{\omega}, \Sigma_{\boldsymbol{x}} + A_{\omega}\Sigma_{\boldsymbol{z}}A_{\omega}^T)\Phi_{\omega}(\boldsymbol{\mu}_{\omega}(\boldsymbol{x}), \Sigma_{\omega})}.$$

$\square$

### F.4 Proof of Lemma 2

The proof will consist of observing that the posterior (prior rewriting) can be expressed as a softmax of a quantity rescaled by the standard deviation.

*Proof.*

$$\begin{aligned}
\log(\phi(\boldsymbol{z}; \boldsymbol{\mu}_{\omega}(\boldsymbol{x}), \Sigma_{\omega})) &= -\frac{1}{2}(\boldsymbol{x} - \boldsymbol{\mu}_{\omega})^T \Sigma_{\omega}^{-1}(\boldsymbol{x} - \boldsymbol{\mu}_{\omega}(\boldsymbol{x}))) - \frac{1}{2}\log(\det(\Sigma_{\omega})) + cst \\
&= -\frac{1}{2}(\boldsymbol{z} - \Sigma_{\omega}A_{\omega}^T\Sigma_{\boldsymbol{x}}^{-1}(A_0\boldsymbol{z}_0 + B_0 - B_{\omega}))^T \Sigma_{\omega}^{-1}(\boldsymbol{z} - \Sigma_{\omega}A_{\omega}^T\Sigma_{\boldsymbol{x}}^{-1}(A_0\boldsymbol{z}_0 + B_0 - B_{\omega}))^T) \\
&\qquad\qquad - \frac{1}{2}\log(\det(\Sigma_{\omega})) + cst \\
&= -\frac{1}{2}(\boldsymbol{z} - (A_{\omega}^T\Sigma_{\boldsymbol{x}}^{-1}A_{\omega})^{-1}A_{\omega}^T\Sigma_{\boldsymbol{x}}^{-1}(A_0\boldsymbol{z}_0 + B_0 - B_{\omega}))^T \Sigma_{\omega}^{-1}(\boldsymbol{z} - (A_{\omega}^T\Sigma_{\boldsymbol{x}}^{-1}A_{\omega})^{-1}A_{\omega}^T\Sigma_{\boldsymbol{x}}^{-1}(A_0\boldsymbol{z}_0 + B_0 - B_{\omega}))^T) \\
&\qquad\qquad - \frac{1}{2}\log(\det(\Sigma_{\omega})) + cst
\end{aligned}$$

where we used the following result to develop $\boldsymbol{\mu}_{\omega}(\boldsymbol{x})$

$$\begin{aligned}
\Sigma_{\omega} = (A_{\omega}^T\Sigma_{\boldsymbol{x}}^{-1}A_{\omega} + \Sigma_{\boldsymbol{z}}^{-1})^{-1} &= (A_{\omega}^T\Sigma_{\boldsymbol{x}}^{-1}A_{\omega} + (A_{\omega}^T\Sigma_{\boldsymbol{x}}^{-1}A_{\omega})(A_{\omega}^T\Sigma_{\boldsymbol{x}}^{-1}A_{\omega})^{-1}\Sigma_{\boldsymbol{z}}^{-1})^{-1} \\
&= (A_{\omega}^T\Sigma_{\boldsymbol{x}}^{-1}A_{\omega})^{-1}(I + (A_{\omega}^T\Sigma_{\boldsymbol{x}}^{-1}A_{\omega})^{-1}\Sigma_{\boldsymbol{z}}^{-1})^{-1} \\
&= (A_{\omega}^T\Sigma_{\boldsymbol{x}}^{-1}A_{\omega})^{-1} \text{ as } (\Sigma_{\boldsymbol{z}}A_{\omega}^T\Sigma_{\boldsymbol{x}}^{-1}A_{\omega})^{-1} \to \boldsymbol{0}.
\end{aligned}$$

if we are in the same region $\omega$ than $\boldsymbol{z}_0$ then the above becomes

$$\arg\max_{\boldsymbol{z} \in \omega_0} \log(\phi(\boldsymbol{z}; \boldsymbol{\mu}_{\omega}(\boldsymbol{x}), \Sigma_{\omega})) = \arg\max_{\boldsymbol{z} \in \omega_0} -\frac{1}{2}(\boldsymbol{z} - \boldsymbol{z}_0)^T \Sigma_{\omega}^{-1}(\boldsymbol{z} - \boldsymbol{z}_0) = \boldsymbol{z}_0,$$

and since we know that we are in the same region, the argmax $\boldsymbol{z} = \boldsymbol{z}_0$ lies in this region and thus is the maximum of the posterior.

$\square$

### F.5 Proof of Lemma 3

*Proof.* The sign vectors represent the sign of each pre-activation feature maps. The key here is that when changing the sign, the input passes through the knot of the corresponding activation function of that layer. This implies a change in the region in the DGN input space. In fact, without degenerate weights and with nonzero activation functions, a change in any dimension of the sign vector (used to form the per region slope and bias) impact a change in the affine mapping used to map inputs $\boldsymbol{z}$ to outputs $\boldsymbol{x}$. As such, whenever a sign changes, the affine mapping changes, leading to a change of region in the DGN input space. As the sign vector is formed from the DGN input space, and we restrict ourselves to the image of this mapping, there does not exist a sign pattern/configuration not reachable by the DGN (otherwise it would not be in the image of this mapping). Now for the other inclusion, recall that a change in region and thus in per region affine mapping can only occur with a change of pre-activation sign pattern. $\square$

### F.6 Proof of Corollary 2

*Proof.* From the above result, it is clear that the preactivation roots define the boundaries of the regions. Obtaining the hyperplane representation of the region thus simply consists of reexpressing this statement with the explicit pre-activation hyperplanes for all the layers and units, the intersection between layers coming from the subdivision. For additional details please see [25]. $\square$

## F.7 Proof of Lemma 4

*Proof.* The proof consists of rearranging the terms from the inclusion-exclusion formula as in

$$\sum_{J\subseteq\{1,...,F\},J\neq\emptyset}(-1)^{|J|+1}\left(\cap_{j\in J}A_j\right)=\cup_i A_i$$

$$(-1)^{F+1}S+\sum_{J\subseteq\{1,...,F\},J\neq\emptyset,|J|<F}(-1)^{|J|+1}\left(\cap_{j\in J}A_j\right)=\cup_i A_i$$

$$(-1)^{F+1}S=\cup_i A_i-\sum_{J\subseteq\{1,...,F\},J\neq\emptyset,|J|<F}(-1)^{|J|+1}\left(\cap_{j\in J}A_j\right)$$

$$S=(-1)^{F+1}\cup_i A_i-(-1)^{F+1}\sum_{J\subseteq\{1,...,F\},J\neq\emptyset,|J|<F}(-1)^{|J|+1}\left(\cap_{j\in J}A_j\right)$$

$$S=(-1)^{F+1}\cup_i A_i+\sum_{J\subseteq\{1,...,F\},J\neq\emptyset,|J|<F}(-1)^{|J|+1+F}\left(\cap_{j\in J}A_j\right)$$

then by application of Chasles rule [64], the integral domain can be decomposed into the signed sum of per cone integration. Finally, a simplex in dimension $S$ has $S+1$ faces, making $F=S+1$ and leading to the desired result. □

## F.8 Proof of Moments

**Lemma 6.** *The first moments of Gaussian integration on an open rectangle defined by its lower limits $\boldsymbol{a}$ is given by*

$$\int_{\boldsymbol{a}}^{\infty}\boldsymbol{z}\phi(\boldsymbol{0},\boldsymbol{\Sigma})d\boldsymbol{z}=\boldsymbol{\Sigma}F(\boldsymbol{a}),\tag{18}$$

$$\int_{\boldsymbol{a}}^{\infty}\boldsymbol{z}\boldsymbol{z}^T\phi(\boldsymbol{0},\boldsymbol{\Sigma})d\boldsymbol{z}=\Phi_{[\boldsymbol{a},\infty)}(\boldsymbol{0},\boldsymbol{\Sigma})\boldsymbol{\Sigma}+\boldsymbol{\Sigma}\left(G(\boldsymbol{a})+\frac{\boldsymbol{a}\odot F(\boldsymbol{a})-(\boldsymbol{\Sigma}\odot G(\boldsymbol{a}))\boldsymbol{1}}{diag(\boldsymbol{\Sigma})}\right)\boldsymbol{\Sigma}.\tag{19}$$

*where the division is performed elementwise.*

*Proof.* First moment:

$$\int_{\omega}\boldsymbol{z}\phi(\boldsymbol{x};\boldsymbol{0},\Sigma)d\boldsymbol{z}=\int_{\omega}\boldsymbol{z}\frac{e^{-\frac{1}{2}\boldsymbol{z}^T\Sigma^{-1}\boldsymbol{z}}}{(2\pi)^{K/2}|\det(\Sigma)|^{1/2}}d\boldsymbol{z}$$

$$=\sum_{\Delta\in S(\omega)}\sum_{(s,C)\in T(\Delta)}s\int_C\boldsymbol{z}\frac{e^{-\frac{1}{2}(R_C\boldsymbol{z})^T(R_C^T)^{-1}\Sigma_\omega^{-1}R_C^{-1}R_C\boldsymbol{z}}}{(2\pi)^{K/2}|\det(\Sigma_\omega)|^{1/2}}d\boldsymbol{z}$$

$$=\sum_{\Delta\in S(\omega)}\sum_{(s,C)\in T(\Delta)}s\int_{\boldsymbol{l}(C)}R^{-1}\boldsymbol{u}\frac{e^{-\frac{1}{2}\boldsymbol{u}^T(R_C\Sigma_\omega R_C^T)^{-1}\boldsymbol{u}}}{(2\pi)^{K/2}|\det(R_C)||\det(\Sigma_\omega)|^{1/2}}d\boldsymbol{u}$$

$$=\sum_{\Delta\in S(\omega)}\sum_{(s,C)\in T(\Delta)}sR_C^{-1}\int_{\boldsymbol{l}(C)}\boldsymbol{u}\phi(\boldsymbol{u};\boldsymbol{0},R_C\Sigma_\omega R_C^T)d\boldsymbol{u}$$

$$=\sum_{\Delta\in S(\omega)}\sum_{(s,C)\in T(\Delta)}sR_C^{-1}(R_C\Sigma_\omega R_C^T F(\boldsymbol{l}(C))$$

$$=\Sigma_\omega\sum_{\Delta\in S(\omega)}\sum_{(s,C)\in T(\Delta)}sR_C^T F(\boldsymbol{l}(C))$$

Second moment

$$\int_\omega zz^T \phi(x; 0, \Sigma)dz = \int_\omega zz^T \frac{e^{-\frac{1}{2}z^T \Sigma^{-1} z}}{(2\pi)^{K/2}|\det(\Sigma)|^{1/2}}dz$$

$$= \sum_{\Delta \in S(\omega)} \sum_{(s,C)\in T(\Delta)} s \int_C zz^T \frac{e^{-\frac{1}{2}(R_C y)^T (R_C^T)^{-1} \Sigma_\omega^{-1} R_C^{-1} R_C y}}{(2\pi)^{K/2}|\det(\Sigma_\omega)|^{1/2}}dz$$

$$= \sum_{\Delta \in S(\omega)} \sum_{(s,C)\in T(\Delta)} s \int_{l(C)} R_C^{-1} uu^T (R_C^{-1})^T \frac{e^{-\frac{1}{2}u^T (R_C \Sigma_\omega R_C^T)^{-1} u}}{(2\pi)^{K/2}|\det(R_C)||\det(\Sigma_\omega)|^{1/2}}du$$

$$= \sum_{\Delta \in S(\omega)} \sum_{(s,C)\in T(\Delta)} sR_C^{-1} \int_{l(C)} uu^T \phi(u; 0, R_C \Sigma_\omega R_C^T)du(R_C^{-1})^T$$

$$= \sum_{\Delta \in S(\omega-\boldsymbol{\mu}_\omega(x))} \sum_{(s,C)\in T(\Delta)} sR_C^{-1}\Big[\Phi_{[l(C),\infty)}(0, R_C \Sigma_\omega R_C^T)R_C \Sigma_\omega R_C^T$$

$$+ R_C \Sigma_\omega R_C^T \left(\frac{l(C) \odot F(l(C)) + (R_C \Sigma_\omega R_C^T \odot G(l(C)))\mathbf{1}}{\mathrm{diag}(R_C \Sigma_\omega R_C^T)}\right)(R_C \Sigma_\omega R_C^T)^T\Big](R_C^{-1})^T$$

$$= \sum_{\Delta \in S(\omega-\boldsymbol{\mu}_\omega(x))} \sum_{(s,C)\in T(\Delta)} s\Big[\Phi_{[l(C),\infty)}(0, R_C \Sigma_\omega R_C^T)\Sigma_\omega$$

$$+ \Sigma_\omega R_C^T \left(\frac{l(C) \odot F(l(C)) + (R_C \Sigma_\omega R_C^T \odot G(l(C)))\mathbf{1}}{\mathrm{diag}(R_C \Sigma_\omega R_C^T)}\right)R_C \Sigma_\omega\Big]$$

$$= e^0_{\omega-\boldsymbol{\mu}_\omega(x)}\Sigma_\omega + \Sigma_\omega\Big[\sum_{\Delta \in S(\omega-\boldsymbol{\mu}_\omega(x))} \sum_{(s,C)\in T(\Delta)} sR_C^T \left(\frac{l(C) \odot F(l(C)) + (R_C \Sigma_\omega R_C^T \odot G(l(C)))\mathbf{1}}{\mathrm{diag}(R_C \Sigma_\omega R_C^T)}\right)R_C\Big]\Sigma_\omega$$

$$\square$$

## F.9  Proof of Theorem 2

*Proof.* Constant: $R_C = \begin{pmatrix} C^T \\ H^T \Sigma_\omega^{-1} \end{pmatrix}$

$$\int_\omega p(z|x)dz = \alpha_\omega(x)\int_\omega \phi(z; \boldsymbol{\mu}_\omega(x), \Sigma_\omega)dz = \alpha_\omega(x)\int_{\omega-\boldsymbol{\mu}_\omega(x)} \phi(z; 0, \Sigma_\omega)dz = \alpha_\omega(x)e^0_{\omega-\boldsymbol{\mu}_\omega(x)}$$

First moment:

$$\int_\omega zp(z|x)dz = \alpha_\omega(x)\int_\omega z \frac{e^{-\frac{1}{2}(z-\boldsymbol{\mu}_\omega(x))^T \Sigma_\omega^{-1}(z-\boldsymbol{\mu}_\omega(x))}}{(2\pi)^{K/2}|\det(\Sigma_\omega)|^{1/2}}dz$$

$$= \alpha_\omega(x)\int_{\omega-\boldsymbol{\mu}_\omega(x)}(y + \boldsymbol{\mu}_\omega(x)) \frac{e^{-\frac{1}{2}y^T \Sigma_\omega^{-1} y}}{(2\pi)^{K/2}|\det(\Sigma_\omega)|^{1/2}}dz$$

$$= \alpha_\omega(x)\left(e^1_{\omega-\boldsymbol{\mu}_\omega(x)} + e^0_{\omega-\boldsymbol{\mu}_\omega(x)}\boldsymbol{\mu}_\omega(x)\right)$$

Second moment:

$$\int zz^T p(z|x)dz = \alpha_\omega(x)\int_\omega zz^T \phi(z; \boldsymbol{\mu}_\omega(x), \Sigma_\omega)dz$$

$$=\alpha_\omega(\boldsymbol{x})\int_{\omega-\mu_\omega(\boldsymbol{x})}(\boldsymbol{y}+\mu_\omega(\boldsymbol{x}))(\boldsymbol{y}+\mu_\omega(\boldsymbol{x}))^T\phi(\boldsymbol{z};\boldsymbol{0},\Sigma_\omega)d\boldsymbol{z}$$

$$=\alpha_\omega(\boldsymbol{x})\left(\boldsymbol{E}^2+\boldsymbol{\mu}_\omega(\boldsymbol{x})\boldsymbol{e}_\omega^1(\boldsymbol{\Sigma}_\omega)^T+\boldsymbol{e}_\omega^1(\boldsymbol{\Sigma}_\omega)\mu_\omega(\boldsymbol{x})^T+\mu_\omega(\boldsymbol{x})\mu_\omega(\boldsymbol{x})^Te_{\omega-\mu_\omega(\boldsymbol{x})}^0(\boldsymbol{\Sigma}_\omega)\right)$$

□

# G  Proof of EM-step

We now derive the expectation maximization steps for a piecewise affine and continuous DGN.

## G.1  E-step derivation

$$E_{\boldsymbol{z}|\boldsymbol{x}}[(A_\omega\boldsymbol{z}+B_\omega)\mathbb{1}_\omega]=Am_\omega^1+Be_\omega^0 \tag{20}$$

$$E_{\boldsymbol{z}|\boldsymbol{x}}[\boldsymbol{z}^TA_\omega^TA_\omega\boldsymbol{z}\mathbb{1}_\omega]=\mathrm{Tr}(A_\omega^TA_\omega m^2) \tag{21}$$

$$E_{Z|X}\left[\log\left(p_{X|Z}(\boldsymbol{x}|\boldsymbol{z})p_Z(\boldsymbol{z})\right)\right]=E_{Z|X}\left[\log\left(\frac{e^{-\frac{1}{2}(\boldsymbol{x}-g(\boldsymbol{z}))^T\Sigma_{\boldsymbol{x}}^{-1}(\boldsymbol{x}-g(\boldsymbol{z}))}}{(2\pi)^{D/2}\sqrt{|\det(\boldsymbol{\Sigma_x})|}}\frac{e^{-\frac{1}{2}\boldsymbol{z}^T\Sigma_{\boldsymbol{z}}^{-1}\boldsymbol{x}}}{(2\pi)^{S/2}\sqrt{|\det(\boldsymbol{\Sigma_z})|}}\right)\right]$$

$$=-\log\left((2\pi)^{(S+D)/2}\sqrt{|\det(\boldsymbol{\Sigma_z})|}\sqrt{|\det(\boldsymbol{\Sigma_x})|}\right)-\frac{1}{2}E_{Z|X}\left[(\boldsymbol{x}-g(\boldsymbol{z}))^T\boldsymbol{\Sigma}_{\boldsymbol{x}}^{-1}(\boldsymbol{x}-g(\boldsymbol{z}))+\boldsymbol{z}^T\boldsymbol{\Sigma}_{\boldsymbol{z}}^{-1}\boldsymbol{z}\right]$$

$$=-\log\left((2\pi)^{(S+D)/2}\sqrt{|\det(\boldsymbol{\Sigma_z})|}\sqrt{|\det(\boldsymbol{\Sigma_x})|}\right)$$
$$-\frac{1}{2}\left(\boldsymbol{x}^T\Sigma_{\boldsymbol{x}}^{-1}\boldsymbol{x}+E_{Z|X}\left[-2\boldsymbol{x}^T\boldsymbol{\Sigma}_{\boldsymbol{x}}^{-1}g(\boldsymbol{z})+g(\boldsymbol{z})^T\boldsymbol{\Sigma}_{\boldsymbol{x}}^{-1}g(\boldsymbol{z})+\boldsymbol{z}^T\boldsymbol{\Sigma}_{\boldsymbol{z}}^{-1}\boldsymbol{z}\right]\right)$$

$$=-\log\left((2\pi)^{(S+D)/2}\sqrt{|\det(\boldsymbol{\Sigma_z})|}\sqrt{|\det(\boldsymbol{\Sigma_x})|}\right)-\frac{1}{2}\left(\boldsymbol{x}^T\Sigma_{\boldsymbol{x}}^{-1}\boldsymbol{x}+\mathrm{Tr}(E_{Z|X}[\boldsymbol{z}\boldsymbol{z}^T\boldsymbol{\Sigma}_{\boldsymbol{z}}^{-1}])\right.$$
$$+E_{Z|X}\left[-2\boldsymbol{x}^T\boldsymbol{\Sigma}_{\boldsymbol{x}}^{-1}g(\boldsymbol{z})+g(\boldsymbol{z})^T\boldsymbol{\Sigma}_{\boldsymbol{x}}^{-1}g(\boldsymbol{z})\right]\bigg)$$

$$=-\log\left((2\pi)^{(S+D)/2}\sqrt{|\det(\boldsymbol{\Sigma_z})|}\sqrt{|\det(\boldsymbol{\Sigma_x})|}\right)-\frac{1}{2}\left(\boldsymbol{x}^T\Sigma_{\boldsymbol{x}}^{-1}\boldsymbol{x}-2\boldsymbol{x}^T\boldsymbol{\Sigma}_{\boldsymbol{x}}^{-1}\left(\sum_\omega\boldsymbol{A}_\omega\boldsymbol{e}_\omega^1(\boldsymbol{x})+\boldsymbol{b}_\omega e_\omega^0(\boldsymbol{x})\right)\right.$$
$$+\sum_\omega e_\omega^0\boldsymbol{b}_\omega^T\boldsymbol{\Sigma}_{\boldsymbol{x}}^{-1}\boldsymbol{b}_\omega+\mathrm{Tr}(A_\omega^T\Sigma_{\boldsymbol{x}}^{-1}A_\omega\boldsymbol{E}_\omega^2(\boldsymbol{x}))+2(\boldsymbol{A}_\omega\boldsymbol{m}_\omega^1(\boldsymbol{x}))^T\boldsymbol{\Sigma}_{\boldsymbol{x}}^{-1}\boldsymbol{b}_\omega+\mathrm{Tr}(\Sigma_{\boldsymbol{z}}^{-1}\boldsymbol{E}^2(\boldsymbol{x}))\bigg)$$

## G.2  Proof of M step

Let first introduce some notations:

$$\boldsymbol{A}_\omega^{L\to i}\triangleq(\boldsymbol{A}_\omega^{L\to i})^T(\text{back-propagation matrix to layer }i),$$

$$r_\omega^\ell(\boldsymbol{x})\triangleq\left(\boldsymbol{x}e_\omega^0(\boldsymbol{x})-\left(\boldsymbol{A}_\omega\boldsymbol{e}_\omega^1(\boldsymbol{x})+\sum_{i\neq\ell}m_\omega^0(\boldsymbol{x})\boldsymbol{A}_\omega^{i+1\to L}\boldsymbol{D}_\omega^i\boldsymbol{v}^i\right)\right)\quad(\text{expected residual without }\boldsymbol{v}^\ell)$$

$$\hat{\boldsymbol{z}}_\omega^\ell(\boldsymbol{x})\triangleq\boldsymbol{D}_\omega^{\ell-1}\left(\boldsymbol{A}_\omega^{1\to\ell-1}m_\omega^1(\boldsymbol{x})+\boldsymbol{b}_\omega^{1\to\ell-1}e_\omega^0\right)\quad(\text{expected feature map of layer }\ell)$$

we can now provide the analytical forms of the M step for each of the learnable parameters:

$$\boldsymbol{\Sigma}_{\boldsymbol{x}}^*=\frac{1}{N}\sum_{\boldsymbol{x}}\left(\boldsymbol{x}\boldsymbol{x}^T+\sum_\omega\boldsymbol{b}_\omega\left(\boldsymbol{b}_\omega m_\omega^0(\boldsymbol{x})+2\boldsymbol{A}_\omega\boldsymbol{e}_\omega^1(\boldsymbol{x})\right)^T-2\boldsymbol{x}(\hat{\boldsymbol{z}}_\omega^L(\boldsymbol{x}))^T+\boldsymbol{A}_\omega\boldsymbol{E}_\omega^2(\boldsymbol{x})\boldsymbol{A}_\omega^T\right), \tag{22}$$

$$\boldsymbol{v}^{\ell *} = \left( \sum_{\boldsymbol{x}} \sum_{\omega} \boldsymbol{D}_\omega^\ell \boldsymbol{A}_\omega^{L\to\ell+1} \boldsymbol{\Sigma}_{\boldsymbol{x}}^{-1} \boldsymbol{A}_\omega^{\ell+1\to L} \boldsymbol{D}_\omega^\ell \right)^{-1} \left( \sum_{\boldsymbol{x}} \sum_{\omega\in\Omega} \underbrace{\boldsymbol{D}_\omega^\ell \boldsymbol{A}_\omega^{L\to\ell+1} \boldsymbol{\Sigma}_{\boldsymbol{x}}^{-1} r_\omega^\ell(\boldsymbol{x})}_{\text{residual back-propagated to layer } \ell} \right), \tag{23}$$

$$\text{vect}(\boldsymbol{W}^{\ell *}) = U_\omega^{-1}\text{vect}\left( \sum_{\boldsymbol{x}} \sum_{\omega} \underbrace{\boldsymbol{D}_\omega^\ell \boldsymbol{A}^{L\to\ell+1} \boldsymbol{\Sigma}_{\boldsymbol{x}}^{-1} \left( \boldsymbol{x} - \sum_{i=\ell}^{L} \boldsymbol{A}_\omega^{i+1\to L} \boldsymbol{D}_\omega^i \boldsymbol{v}^i \right)}_{\text{residual back-propagated to layer } \ell} (\hat{\boldsymbol{z}}_\omega^\ell(\boldsymbol{x}))^T \right), \tag{24}$$

we provide detailed derivations below.

### G.2.1 Update of the bias parameter

Recall from (2) that $\boldsymbol{b}_\omega = \boldsymbol{v}^L + \sum_{i=1}^{L-1} \boldsymbol{W}^L \boldsymbol{D}_\omega^{L-1} \boldsymbol{W}^{L-1} \dots \boldsymbol{D}_\omega^i \boldsymbol{v}^i$, we can thus rewrite the loss as

$$L(\boldsymbol{v}^\ell) = -\frac{1}{2} \log\left( (2\pi)^{S+D} |\det(\boldsymbol{\Sigma}_{\boldsymbol{x}})|| \det(\boldsymbol{\Sigma}_{\boldsymbol{z}})| \right) - \frac{1}{2}\left( \boldsymbol{x}^T \boldsymbol{\Sigma}_{\boldsymbol{x}}^{-1} \boldsymbol{x} - 2\boldsymbol{x}^T \boldsymbol{\Sigma}_{\boldsymbol{x}}^{-1} \left( \sum_\omega \boldsymbol{A}_\omega m_\omega^1(\boldsymbol{x}) + \boldsymbol{b}_\omega m_\omega^0(\boldsymbol{x}) \right) \right.$$

$$\left. + \sum_\omega m_\omega^0 \boldsymbol{b}_\omega^T \boldsymbol{\Sigma}_{\boldsymbol{x}}^{-1} \boldsymbol{b}_\omega + \text{Tr}(\boldsymbol{A}_\omega^T \boldsymbol{\Sigma}_{\boldsymbol{x}}^{-1} \boldsymbol{A}_\omega M_\omega^2(\boldsymbol{x})) + 2(\boldsymbol{A}_\omega m_\omega^1(\boldsymbol{x}))^T \boldsymbol{\Sigma}_{\boldsymbol{x}}^{-1} \boldsymbol{b}_\omega \right) - \frac{1}{2}\text{Tr}(\boldsymbol{\Sigma}_{\boldsymbol{z}}^{-1} M^2(\boldsymbol{x}))$$

$$= -\frac{1}{2}\left( -2\boldsymbol{x}^T \boldsymbol{\Sigma}_{\boldsymbol{x}}^{-1} \left( \sum_\omega \boldsymbol{b}_\omega e_\omega^0(\boldsymbol{x}) \right) + \sum_\omega e_\omega^0 \boldsymbol{b}_\omega^T \boldsymbol{\Sigma}_{\boldsymbol{x}}^{-1} \boldsymbol{b}_\omega + 2\sum_\omega (\boldsymbol{A}_\omega m_\omega^1(\boldsymbol{x}))^T \boldsymbol{\Sigma}_{\boldsymbol{x}}^{-1} \boldsymbol{b}_\omega \right) + cst$$

$$= -\frac{1}{2}\sum_\omega \left( -2\boldsymbol{x}^T \boldsymbol{\Sigma}_{\boldsymbol{x}}^{-1} (\boldsymbol{A}_\omega^{\ell+1\to L} \boldsymbol{D}_\omega^\ell \boldsymbol{v}^\ell e_\omega^0(\boldsymbol{x})) + e_\omega^0 (\boldsymbol{A}_\omega^{\ell+1\to L} \boldsymbol{D}_\omega^\ell \boldsymbol{v}^\ell)^T \boldsymbol{\Sigma}_{\boldsymbol{x}}^{-1} (\boldsymbol{A}_\omega^{\ell+1\to L} \boldsymbol{D}_\omega^\ell \boldsymbol{v}^\ell) \right.$$

$$\left. + 2e_\omega^0(\boldsymbol{x})(\sum_{i\neq\ell} \boldsymbol{A}_\omega^{i+1\to L} \boldsymbol{D}_\omega^i \boldsymbol{v}^i)^T \boldsymbol{\Sigma}_{\boldsymbol{x}}^{-1} (\boldsymbol{A}_\omega^{\ell+1\to L} \boldsymbol{D}_\omega^\ell \boldsymbol{v}^\ell) + 2((m_\omega^1(\boldsymbol{x}))^T (\boldsymbol{A}_\omega)^T \boldsymbol{\Sigma}_{\boldsymbol{x}}^{-1} \boldsymbol{A}_\omega^{\ell+1\to L} \boldsymbol{D}_\omega^\ell \boldsymbol{v}^\ell \right) + cst$$

$$= -\frac{1}{2}\sum_\omega \left( e_\omega^0 (\boldsymbol{A}_\omega^{\ell+1\to L} \boldsymbol{D}_\omega^\ell \boldsymbol{v}^\ell)^T \boldsymbol{\Sigma}_{\boldsymbol{x}}^{-1} (\boldsymbol{A}_\omega^{\ell+1\to L} \boldsymbol{D}_\omega^\ell \boldsymbol{v}^\ell) \right. \tag{A}$$

$$\left. + 2(e_\omega^0(\boldsymbol{x})(\sum_{i\neq\ell} \boldsymbol{A}_\omega^{i+1\to L} \boldsymbol{D}_\omega^i \boldsymbol{v}^i - \boldsymbol{x}) + \boldsymbol{A}_\omega e_\omega^1(\boldsymbol{x}))^T \boldsymbol{\Sigma}_{\boldsymbol{x}}^{-1} (\boldsymbol{A}_\omega^{\ell+1\to L} \boldsymbol{D}_\omega^\ell \boldsymbol{v}^\ell) \right) + cst \tag{B}$$

$$\implies \partial L(\boldsymbol{v}^\ell) = -\frac{1}{2}\sum_\omega \left[ -e_\omega^0(\boldsymbol{x})2\boldsymbol{D}_\omega^\ell \boldsymbol{A}_\omega^{L\to\ell+1} \boldsymbol{\Sigma}_{\boldsymbol{x}}^{-1} \boldsymbol{A}_\omega^{\ell+1\to L} \boldsymbol{D}_\omega^\ell \boldsymbol{v}^\ell \right.$$

$$\left. + 2\left( \boldsymbol{A}_\omega^{\ell+1\to L} \boldsymbol{D}_\omega^\ell \right)^T \boldsymbol{\Sigma}_{\boldsymbol{x}}^{-1} \left( e_\omega^0(\boldsymbol{x}) \left( \sum_{i\neq\ell} \boldsymbol{A}_\omega^{i+1\to L} \boldsymbol{D}_\omega^i \boldsymbol{v}^i - \boldsymbol{x} \right) + \boldsymbol{A}_\omega e_\omega^1(\boldsymbol{x}) \right) \right]$$

$$\implies \boldsymbol{v}^\ell = \left( \sum_{\boldsymbol{x}} \sum_\omega e_\omega^0(\boldsymbol{x}) \boldsymbol{D}_\omega^\ell \boldsymbol{A}_\omega^{L\to\ell+1} \boldsymbol{\Sigma}_{\boldsymbol{x}}^{-1} \boldsymbol{A}_\omega^{\ell+1\to L} \boldsymbol{D}_\omega^\ell \right)^{-1}$$

$$\times \sum_{\boldsymbol{x}} \sum_{\omega\in\Omega} \boldsymbol{D}_\omega^\ell \boldsymbol{A}_\omega^{L\to\ell+1} \boldsymbol{\Sigma}_{\boldsymbol{x}}^{-1} \left( \boldsymbol{x} e_\omega^0(\boldsymbol{x}) - \left( \boldsymbol{A}_\omega m_\omega^1(\boldsymbol{x}) + \sum_{i\neq\ell} m_\omega^0(\boldsymbol{x}) \boldsymbol{A}_\omega^{i+1\to L} \boldsymbol{D}_\omega^i \boldsymbol{v}^i \right) \right)$$

as

$$(A) = e_\omega^0(\boldsymbol{x})(\boldsymbol{A}_\omega^{\ell+1\to L}\boldsymbol{D}_\omega^\ell\boldsymbol{v}^\ell)^T\boldsymbol{\Sigma}_{\boldsymbol{x}}^{-1}(\boldsymbol{A}_\omega^{\ell+1\to L}\boldsymbol{D}_\omega^\ell\boldsymbol{v}^\ell)$$

$$\implies \partial(A) = e_\omega^0(\boldsymbol{x})2\boldsymbol{D}_\omega^\ell\boldsymbol{A}_\omega^{L\to\ell+1}\boldsymbol{\Sigma}_{\boldsymbol{x}}^{-1}\boldsymbol{A}_\omega^{\ell+1\to L}\boldsymbol{D}_\omega^\ell\boldsymbol{v}^\ell$$

$$(B) = 2\left[\left(e_\omega^0(\boldsymbol{x})\left(\sum_{i\neq\ell}\boldsymbol{A}_\omega^{i+1\to L}\boldsymbol{D}_\omega^i\boldsymbol{v}^i - \boldsymbol{x}\right) + \boldsymbol{A}_\omega e_\omega^1(\boldsymbol{x})\right)^T\boldsymbol{\Sigma}_{\boldsymbol{x}}^{-1}(\boldsymbol{A}_\omega^{\ell+1\to L}\boldsymbol{D}_\omega^\ell\boldsymbol{v}^\ell)\right] + cst$$

$$\implies \partial(B) = \left(\boldsymbol{A}_\omega^{\ell+1\to L}\boldsymbol{D}_\omega^\ell\right)^T\boldsymbol{\Sigma}_{\boldsymbol{x}}^{-1}\left(e_\omega^0(\boldsymbol{x})\left(\sum_{i\neq\ell}\boldsymbol{A}_\omega^{i+1\to L}\boldsymbol{D}_\omega^i\boldsymbol{v}^i - \boldsymbol{x}\right) + \boldsymbol{A}_\omega e_\omega^1(\boldsymbol{x})\right)$$

### G.2.2 Update of the slope parameter

We can thus rewrite the loss as

$$L(\boldsymbol{v}^\ell) = -\frac{1}{2}\log\left((2\pi)^{S+D}|\det(\boldsymbol{\Sigma}_{\boldsymbol{x}})||\det(\boldsymbol{\Sigma}_{\boldsymbol{z}})|\right) - \frac{1}{2}\left(\boldsymbol{x}^T\boldsymbol{\Sigma}_{\boldsymbol{x}}^{-1}\boldsymbol{x} - 2\boldsymbol{x}^T\boldsymbol{\Sigma}_{\boldsymbol{x}}^{-1}\left(\sum_\omega\boldsymbol{A}_\omega m_\omega^1(\boldsymbol{x}) + \boldsymbol{b}_\omega m_\omega^0(\boldsymbol{x})\right)\right.$$

$$\left.+ \sum_\omega m_\omega^0\boldsymbol{b}_\omega^T\boldsymbol{\Sigma}_{\boldsymbol{x}}^{-1}\boldsymbol{b}_\omega + \text{Tr}(\boldsymbol{A}_\omega^T\boldsymbol{\Sigma}_{\boldsymbol{x}}^{-1}\boldsymbol{A}_\omega\boldsymbol{M}_\omega^2(\boldsymbol{x})) + 2(\boldsymbol{A}_\omega m_\omega^1(\boldsymbol{x}))^T\boldsymbol{\Sigma}_{\boldsymbol{x}}^{-1}\boldsymbol{b}_\omega\right) - \frac{1}{2}\text{Tr}(\boldsymbol{\Sigma}_{\boldsymbol{z}}^{-1}\boldsymbol{M}^2(\boldsymbol{x}))$$

$$= \boldsymbol{x}^T\boldsymbol{\Sigma}_{\boldsymbol{x}}^{-1}\left(\sum_\omega\boldsymbol{A}_\omega e_\omega^1(\boldsymbol{x}) + \boldsymbol{b}_\omega e_\omega^0(\boldsymbol{x})\right) - \frac{1}{2}\sum_\omega e_\omega^0\boldsymbol{b}_\omega^T\boldsymbol{\Sigma}_{\boldsymbol{x}}^{-1}\boldsymbol{b}_\omega - \frac{1}{2}\sum_\omega\text{Tr}(\boldsymbol{A}_\omega^T\boldsymbol{\Sigma}_{\boldsymbol{x}}^{-1}\boldsymbol{A}_\omega\boldsymbol{E}_\omega^2(\boldsymbol{x}))$$

$$- \sum_\omega(\boldsymbol{A}_\omega m_\omega^1(\boldsymbol{x}))^T\boldsymbol{\Sigma}_{\boldsymbol{x}}^{-1}\boldsymbol{b}_\omega$$

Notice that we can rewrite $\boldsymbol{b}_\omega = \boldsymbol{A}_\omega^{\ell+1\to L}\boldsymbol{D}_\omega^\ell\boldsymbol{W}^\ell\boldsymbol{D}_\omega^{\ell-1}\boldsymbol{b}_\omega^{1\to\ell-1} + \sum_{i=\ell}^L\boldsymbol{A}_\omega^{i+1\to L}\boldsymbol{D}_\omega^i\boldsymbol{v}^i$ and $\boldsymbol{A}_\omega = \boldsymbol{A}_\omega^{\ell+1\to L}\boldsymbol{D}_\omega^\ell\boldsymbol{W}^\ell\boldsymbol{D}_\omega^{\ell-1}\boldsymbol{A}_\omega^{1\to\ell-1}$ and thus we obtain:

$$L(\boldsymbol{v}^\ell) = \sum_\omega\boldsymbol{x}^T\boldsymbol{\Sigma}_{\boldsymbol{x}}^{-1}\boldsymbol{A}_\omega^{\ell+1\to L}\boldsymbol{D}_\omega^\ell\boldsymbol{W}^\ell\boldsymbol{D}_\omega^{\ell-1}\left(\boldsymbol{A}^{1\to\ell-1}e_\omega^1(\boldsymbol{x}) + \boldsymbol{b}_\omega^{1\to\ell-1}e_\omega^0(\boldsymbol{x})\right)$$

$$- \frac{1}{2}\sum_\omega e_\omega^0(\boldsymbol{A}^{\ell+1\to L}\boldsymbol{D}_\omega^\ell\boldsymbol{W}^\ell\boldsymbol{D}_\omega^{\ell-1}\boldsymbol{b}_\omega^{1\to\ell-1})^T\boldsymbol{\Sigma}_{\boldsymbol{x}}^{-1}(\boldsymbol{A}^{\ell+1\to L}\boldsymbol{D}_\omega^\ell\boldsymbol{W}^\ell\boldsymbol{D}_\omega^{\ell-1}\boldsymbol{b}_\omega^{1\to\ell-1})$$

$$- \sum_\omega e_\omega^0(\boldsymbol{A}^{\ell+1\to L}\boldsymbol{D}_\omega^\ell\boldsymbol{W}^\ell\boldsymbol{D}_\omega^{\ell-1}\boldsymbol{b}_\omega^{1\to\ell-1})^T\boldsymbol{\Sigma}_{\boldsymbol{x}}^{-1}(\sum_{i=\ell}^L\boldsymbol{A}_\omega^{i+1\to L}\boldsymbol{D}_\omega^i\boldsymbol{v}^i)$$

$$- \frac{1}{2}\sum_\omega\text{Tr}((\boldsymbol{A}_\omega^{\ell+1\to L}\boldsymbol{D}_\omega^\ell\boldsymbol{W}^\ell\boldsymbol{D}_\omega^{\ell-1}\boldsymbol{A}^{1\to\ell-1})^T\boldsymbol{\Sigma}_{\boldsymbol{x}}^{-1}(\boldsymbol{A}_\omega^{\ell+1\to L}\boldsymbol{D}_\omega^\ell\boldsymbol{W}^\ell\boldsymbol{D}_\omega^{\ell-1}\boldsymbol{A}^{1\to\ell-1})\boldsymbol{E}_\omega^2(\boldsymbol{x}))$$

$$- \sum_\omega(\boldsymbol{A}_\omega^{\ell+1\to L}\boldsymbol{D}_\omega^\ell\boldsymbol{W}^\ell\boldsymbol{D}_\omega^{\ell-1}\boldsymbol{A}^{1\to\ell-1}m_\omega^1(\boldsymbol{x}))^T\boldsymbol{\Sigma}_{\boldsymbol{x}}^{-1}(\boldsymbol{A}^{\ell+1\to L}\boldsymbol{D}_\omega^\ell\boldsymbol{W}^\ell\boldsymbol{D}_\omega^{\ell-1}\boldsymbol{b}_\omega^{1\to\ell-1})$$

$$- \sum_\omega(\boldsymbol{A}_\omega^{\ell+1\to L}\boldsymbol{D}_\omega^\ell\boldsymbol{W}^\ell\boldsymbol{D}_\omega^{\ell-1}\boldsymbol{A}^{1\to\ell-1}m_\omega^1(\boldsymbol{x}))^T\boldsymbol{\Sigma}_{\boldsymbol{x}}^{-1}(\sum_{i=\ell}^L\boldsymbol{A}_\omega^{i+1\to L}\boldsymbol{D}_\omega^i\boldsymbol{v}^i) + cst$$

$$= \sum_\omega\boldsymbol{x}^T\boldsymbol{\Sigma}_{\boldsymbol{x}}^{-1}\boldsymbol{A}^{\ell+1\to L}\boldsymbol{D}_\omega^\ell\boldsymbol{W}^\ell\boldsymbol{D}_\omega^{\ell-1}\left(\boldsymbol{A}^{1\to\ell-1}e_\omega^1(\boldsymbol{x}) + \boldsymbol{b}_\omega^{1\to\ell-1}e_\omega^0(\boldsymbol{x})\right) \tag{A}$$

$$-\frac{1}{2}\sum_\omega e_\omega^0 (\boldsymbol{A}^{\ell+1\to L}\boldsymbol{D}_\omega^\ell \boldsymbol{W}^\ell \boldsymbol{D}_\omega^{\ell-1}\boldsymbol{b}_\omega^{1\to\ell-1})^T \boldsymbol{\Sigma}_{\boldsymbol{x}}^{-1}(\boldsymbol{A}^{\ell+1\to L}\boldsymbol{D}_\omega^\ell \boldsymbol{W}^\ell \boldsymbol{D}_\omega^{\ell-1}\boldsymbol{b}_\omega^{1\to\ell-1}) \qquad (B)$$

$$-\sum_\omega \Big(\boldsymbol{A}^{\ell+1\to L}\boldsymbol{D}_\omega^\ell \boldsymbol{W}^\ell \boldsymbol{D}_\omega^{\ell-1}\big(\boldsymbol{A}^{1\to\ell-1}\boldsymbol{e}_\omega^1(\boldsymbol{x}) + \boldsymbol{b}_\omega^{1\to\ell-1}e_\omega^0(\boldsymbol{x})\big)\Big)^T \boldsymbol{\Sigma}_{\boldsymbol{x}}^{-1}(\sum_{i=\ell}^{L}\boldsymbol{A}_\omega^{i+1\to L}\boldsymbol{D}_\omega^i \boldsymbol{v}^i) \qquad (C)$$

$$-\frac{1}{2}\sum_\omega \mathrm{Tr}((\boldsymbol{A}_\omega^{\ell+1\to L}\boldsymbol{D}_\omega^\ell \boldsymbol{W}^\ell \boldsymbol{D}_\omega^{\ell-1}\boldsymbol{A}^{1\to\ell-1})^T \boldsymbol{\Sigma}_{\boldsymbol{x}}^{-1}(\boldsymbol{A}_\omega^{\ell+1\to L}\boldsymbol{D}_\omega^\ell \boldsymbol{W}^\ell \boldsymbol{D}_\omega^{\ell-1}\boldsymbol{A}^{1\to\ell-1})\boldsymbol{E}_\omega^2(\boldsymbol{x})) \qquad (D)$$

$$-\sum_\omega (\boldsymbol{A}_\omega^{\ell+1\to L}\boldsymbol{D}_\omega^\ell \boldsymbol{W}^\ell \boldsymbol{D}_\omega^{\ell-1}\boldsymbol{A}^{1\to\ell-1}m_\omega^1(\boldsymbol{x}))^T \boldsymbol{\Sigma}_{\boldsymbol{x}}^{-1}(\boldsymbol{A}_\omega^{\ell+1\to L}\boldsymbol{D}_\omega^\ell \boldsymbol{W}^\ell \boldsymbol{D}_\omega^{\ell-1}\boldsymbol{b}_\omega^{1\to\ell-1}) + cst \qquad (E)$$

$$A = \sum_\omega \boldsymbol{x}^T \boldsymbol{\Sigma}_{\boldsymbol{x}}^{-1}\boldsymbol{A}^{\ell+1\to L}\boldsymbol{D}_\omega^\ell \boldsymbol{W}^\ell \boldsymbol{D}_\omega^{\ell-1}\Big(\boldsymbol{A}^{1\to\ell-1}\boldsymbol{e}_\omega^1(\boldsymbol{x}) + \boldsymbol{b}_\omega^{1\to\ell-1}e_\omega^0(\boldsymbol{x})\Big)$$

$$\implies \partial A = \sum_\omega \boldsymbol{D}_\omega^\ell \boldsymbol{A}_\omega^{L\to\ell+1}\boldsymbol{\Sigma}_{\boldsymbol{x}}^{-1}\boldsymbol{x}\left(\boldsymbol{D}_\omega^{\ell-1}(\boldsymbol{A}^{1\to\ell-1}\boldsymbol{e}_\omega^1(\boldsymbol{x}) + \boldsymbol{b}^{1\to\ell-1}e_\omega^0(\boldsymbol{x}))\right)^T$$

$$B = -\frac{1}{2}\sum_\omega e_\omega^0(\boldsymbol{x})\left(\boldsymbol{A}^{\ell+1\to L}\boldsymbol{D}_\omega^\ell \boldsymbol{W}^\ell \boldsymbol{D}_\omega^{\ell-1}\boldsymbol{b}_\omega^{1\to\ell-1}\right)^T \boldsymbol{\Sigma}_{\boldsymbol{x}}^{-1}\left(\boldsymbol{A}^{\ell+1\to L}\boldsymbol{D}_\omega^\ell \boldsymbol{W}^\ell \boldsymbol{D}_\omega^{\ell-1}\boldsymbol{b}_\omega^{1\to\ell-1}\right)$$

$$= -\frac{1}{2}\sum_\omega e_\omega^0(\boldsymbol{x})(\boldsymbol{D}_\omega^{\ell-1}\boldsymbol{b}_\omega^{1\to\ell-1})^T (\boldsymbol{W}^\ell)^T (\boldsymbol{A}^{\ell+1\to L}\boldsymbol{D}_\omega^\ell)^T \boldsymbol{\Sigma}_{\boldsymbol{x}}^{-1}\left(\boldsymbol{A}^{\ell+1\to L}\boldsymbol{D}_\omega^\ell \boldsymbol{W}^\ell \boldsymbol{D}_\omega^{\ell-1}\boldsymbol{b}_\omega^{1\to\ell-1}\right)$$

$$= -\frac{1}{2}\sum_\omega e_\omega^0(\boldsymbol{x})\mathrm{Tr}\left((\boldsymbol{W}^\ell)^T (\boldsymbol{A}^{\ell+1\to L}\boldsymbol{D}_\omega^\ell)^T \boldsymbol{\Sigma}_{\boldsymbol{x}}^{-1}\boldsymbol{A}^{\ell+1\to L}\boldsymbol{D}_\omega^\ell \boldsymbol{W}^\ell \boldsymbol{D}_\omega^{\ell-1}\boldsymbol{b}_\omega^{1\to\ell-1}(\boldsymbol{D}_\omega^{\ell-1}\boldsymbol{b}_\omega^{1\to\ell-1})^T\right)$$

$$\implies \partial B = -\sum_\omega e_\omega^0(\boldsymbol{x})\boldsymbol{D}_\omega^\ell \boldsymbol{A}^{L\to\ell+1}\boldsymbol{\Sigma}_{\boldsymbol{x}}^{-1}\boldsymbol{A}^{\ell+1\to L}\boldsymbol{D}_\omega^\ell \boldsymbol{W}^\ell \boldsymbol{D}_\omega^{\ell-1}\boldsymbol{b}_\omega^{1\to\ell-1}(\boldsymbol{b}_\omega^{1\to\ell-1})^T \boldsymbol{D}_\omega^{\ell-1}$$

$$C = -\sum_\omega \Big(\boldsymbol{A}^{\ell+1\to L}\boldsymbol{D}_\omega^\ell \boldsymbol{W}^\ell \boldsymbol{D}_\omega^{\ell-1}\big(\boldsymbol{A}^{1\to\ell-1}\boldsymbol{e}_\omega^1(\boldsymbol{x}) + \boldsymbol{b}_\omega^{1\to\ell-1}e_\omega^0(\boldsymbol{x})\big)\Big)^T \boldsymbol{\Sigma}_{\boldsymbol{x}}^{-1}(\sum_{i=\ell}^{L}\boldsymbol{A}_\omega^{i+1\to L}\boldsymbol{D}_\omega^i \boldsymbol{v}^i)$$

$$= -\sum_\omega (\boldsymbol{D}_\omega^{\ell-1}\big(\boldsymbol{A}^{1\to\ell-1}\boldsymbol{e}_\omega^1(\boldsymbol{x}) + \boldsymbol{b}_\omega^{1\to\ell-1}e_\omega^0(\boldsymbol{x})\big)^T (\boldsymbol{W}^\ell)^T (\boldsymbol{A}^{\ell+1\to L}\boldsymbol{D}_\omega^\ell)^T \boldsymbol{\Sigma}_{\boldsymbol{x}}^{-1}(\sum_{i=\ell}^{L}\boldsymbol{A}_\omega^{i+1\to L}\boldsymbol{D}_\omega^i \boldsymbol{v}^i)$$

$$\implies \partial C = -\sum_\omega \boldsymbol{D}_\omega^\ell \boldsymbol{A}^{L\to\ell+1}\boldsymbol{\Sigma}_{\boldsymbol{x}}^{-1}(\sum_{i=\ell}^{L}\boldsymbol{A}_\omega^{i+1\to L}\boldsymbol{D}_\omega^i \boldsymbol{v}^i)(\boldsymbol{D}_\omega^{\ell-1}\big(\boldsymbol{A}^{1\to\ell-1}\boldsymbol{e}_\omega^1(\boldsymbol{x}) + \boldsymbol{b}_\omega^{1\to\ell-1}e_\omega^0(\boldsymbol{x})\big)^T$$

$$D = -\frac{1}{2}\sum_\omega \mathrm{Tr}((\boldsymbol{A}_\omega^{\ell+1\to L}\boldsymbol{D}_\omega^\ell \boldsymbol{W}^\ell \boldsymbol{D}_\omega^{\ell-1}\boldsymbol{A}^{1\to\ell-1})^T \boldsymbol{\Sigma}_{\boldsymbol{x}}^{-1}\boldsymbol{A}_\omega^{\ell+1\to L}\boldsymbol{D}_\omega^\ell \boldsymbol{W}^\ell \boldsymbol{D}_\omega^{\ell-1}\boldsymbol{A}^{1\to\ell-1}\boldsymbol{E}_\omega^2(\boldsymbol{x}))$$

$$= -\frac{1}{2}\sum_\omega \mathrm{Tr}((\boldsymbol{W}^\ell)^T (\boldsymbol{A}^{\ell+1\to L}\boldsymbol{D}_\omega^\ell)^T \boldsymbol{\Sigma}_{\boldsymbol{x}}^{-1}\boldsymbol{A}_\omega^{\ell+1\to L}\boldsymbol{D}_\omega^\ell \boldsymbol{W}^\ell \boldsymbol{D}_\omega^{\ell-1}\boldsymbol{A}^{1\to\ell-1}\boldsymbol{E}_\omega^2(\boldsymbol{x})(\boldsymbol{D}_\omega^{\ell-1}\boldsymbol{A}_\omega^{1\to\ell-1})^T)$$

$$\implies \partial D = -\sum_\omega \boldsymbol{D}_\omega^\ell \boldsymbol{A}^{L\to\ell+1}\boldsymbol{\Sigma}_{\boldsymbol{x}}^{-1}\boldsymbol{A}_\omega^{\ell+1\to L}\boldsymbol{D}_\omega^\ell \boldsymbol{W}^\ell \boldsymbol{D}_\omega^{\ell-1}\boldsymbol{A}^{1\to\ell-1}\boldsymbol{E}_\omega^2(\boldsymbol{x})\boldsymbol{A}_\omega^{\ell-1\to1}\boldsymbol{D}_\omega^{\ell-1}$$

$$E = -\sum_{\omega} \left( \boldsymbol{A}_{\omega}^{\ell+1\to L} \boldsymbol{D}_{\omega}^{\ell} \boldsymbol{W}^{\ell} \boldsymbol{D}_{\omega}^{\ell-1} \boldsymbol{A}^{1\to\ell-1} m_{\omega}^1(\boldsymbol{x}) \right)^T \boldsymbol{\Sigma}_{\boldsymbol{x}}^{-1} \left( \boldsymbol{A}_{\omega}^{\ell+1\to L} \boldsymbol{D}_{\omega}^{\ell} \boldsymbol{W}^{\ell} \boldsymbol{D}_{\omega}^{\ell-1} \boldsymbol{b}_{\omega}^{1\to\ell-1} \right)$$

$$= -\sum_{\omega} \mathrm{Tr}\left( (\boldsymbol{W}^{\ell})^T (\boldsymbol{A}_{\omega}^{\ell+1\to L} \boldsymbol{D}_{\omega}^{\ell})^T \boldsymbol{\Sigma}_{\boldsymbol{x}}^{-1} \boldsymbol{A}^{\ell+1\to L} \boldsymbol{D}_{\omega}^{\ell} \boldsymbol{W}^{\ell} \boldsymbol{D}_{\omega}^{\ell-1} \boldsymbol{b}_{\omega}^{1\to\ell-1} (\boldsymbol{D}_{\omega}^{\ell-1} \boldsymbol{A}^{1\to\ell-1} m_{\omega}^1(\boldsymbol{x}))^T \right)$$

$$\implies \partial E = -\sum_{\omega} \boldsymbol{D}_{\omega}^{\ell} \boldsymbol{A}^{L\to\ell+1} \boldsymbol{\Sigma}_{\boldsymbol{x}}^{-1} \boldsymbol{A}^{\ell+1\to L} \boldsymbol{D}_{\omega}^{\ell} \boldsymbol{W}^{\ell} \left( \boldsymbol{D}^{\ell-1} \left( \boldsymbol{b}_{\omega}^{1\to\ell-1} (m_{\omega}^1(\boldsymbol{x}))^T \boldsymbol{A}_{\omega}^{\ell-1\to 1} + \boldsymbol{A}_{\omega}^{1\to\ell-1} m_{\omega}^1(\boldsymbol{x})(\boldsymbol{b}_{\omega}^{1\to\ell-1})^T \right) (\boldsymbol{D}^{\ell-1})^T \right)$$

we can group B,D and E together as well as A and C. Now to solve this equal 0 we will need to consider the flatten version of $\boldsymbol{W}^{\ell}$ which we denote by $\boldsymbol{w}^{\ell} = \mathrm{vect}(\boldsymbol{W}^{\ell})$ leading to

$$\partial L = \sum_{\omega} \boldsymbol{D}_{\omega}^{\ell} \boldsymbol{A}_{\omega}^{L\to\ell+1} \boldsymbol{\Sigma}_{\boldsymbol{x}}^{-1} (\boldsymbol{x} - \sum_{i=\ell}^{L} \boldsymbol{A}_{\omega}^{i+1\to L} \boldsymbol{D}_{\omega}^{i} \boldsymbol{v}^i) \left( \boldsymbol{D}_{\omega}^{\ell-1} (\boldsymbol{A}^{1\to\ell-1} e_{\omega}^1(\boldsymbol{x}) + \boldsymbol{b}^{1\to\ell-1} e_{\omega}^0(\boldsymbol{x})) \right)^T$$

$$- \sum_{\omega} \boldsymbol{D}_{\omega}^{\ell} \boldsymbol{A}^{L\to\ell+1} \boldsymbol{\Sigma}_{\boldsymbol{x}}^{-1} \boldsymbol{A}_{\omega}^{\ell+1\to L} \boldsymbol{D}_{\omega}^{\ell} \boldsymbol{W}^{\ell} \boldsymbol{D}_{\omega}^{\ell-1} \left( e_{\omega}^0(\boldsymbol{x}) \boldsymbol{b}_{\omega}^{1\to\ell-1} (\boldsymbol{b}_{\omega}^{1\to\ell-1})^T + \boldsymbol{A}_{\omega}^{1\to\ell-1} \boldsymbol{E}_{\omega}^2(\boldsymbol{x}) \boldsymbol{A}_{\omega}^{\ell-1\to 1} \right.$$

$$\left. + \boldsymbol{b}_{\omega}^{1\to\ell-1} (m_{\omega}^1(\boldsymbol{x}))^T \boldsymbol{A}_{\omega}^{\ell-1\to 1} + \boldsymbol{A}_{\omega}^{1\to\ell-1} m_{\omega}^1(\boldsymbol{x})(\boldsymbol{b}_{\omega}^{1\to\ell-1})^T \right) \boldsymbol{D}_{\omega}^{\ell-1}$$

$$= \sum_{\omega} P_{\omega}(\boldsymbol{x})^{\ell} - U_{\omega}^{\ell} \boldsymbol{W}^{\ell} V_{\omega}^{\ell}(\boldsymbol{x})$$

$$\implies (\sum_{\boldsymbol{x}} \sum_{\omega} U_{\omega}^{\ell} \otimes (V_{\omega}^{\ell}(\boldsymbol{x}))^T) \mathrm{vect}(\boldsymbol{W}^{\ell}) = \sum_{\boldsymbol{x}} \sum_{\omega} \mathrm{vect}(P_{\omega}^{\ell}(\boldsymbol{x}))$$

$$\implies \mathrm{vect}(\boldsymbol{W}^{\ell})^* = (\sum_{\boldsymbol{x}} \sum_{\omega} U_{\omega}^{\ell} \otimes (V_{\omega}^{\ell}(\boldsymbol{x}))^T)^{-1} (\sum_{\boldsymbol{x}} \sum_{\omega} \mathrm{vect}(P_{\omega}^{\ell}(\boldsymbol{x})))$$

## H   Regularization

We propose in this section a brief discussion on the impact of using a probabilistic prior on the weights of the GDN. In particular, it is clear that imposing a Gaussian prior with zero mean and isotropic covariance on the weights falls back in the log likelihood to impose a $l2$ regularization of the weights with parameter based on the covariance of the prior. If the prior is a Laplace distribution, the log-likelihood will turn the prior into an $l1$ regularization of the weights, again with regularization coefficient based on the prior covariance. Finally, in the case of uniform prior with finite support, the log likelihood will be equivalent to a weight clipping, a standard technique employed in DNs where the weights can not take values outside of a predefined range.

## I   Computational Complexity

The computational complexity of the method increases drastically with the latent space dimension, and the number of regions, and the number of faces per regions. Those last quantities are directly tied into the complexity (depth and width) of the DGNs. This complexity bottleneck comes from the need to search for all regions, and the need to decompose each region into simplices. As such, the EM learning is not yet suitable for large scale application, however based on the obtained analytical forms, it is possible to derive an approximation of the true form that would be more tractable while providing approximation error bounds as opposed to current methods.

## J   Additional Experiments

In this section we propose to complement the toy circle experiment from the main paper first we an additional $2d$ case and then with the MNIST dataset.

Figure 10: Depiction of the evolution of the NLL during training for the EM and VAE algorithms, we can see that despite the high number of training steps, VAEs are not yet able to correctly approximate the data distribution as opposed to EM training which benefits from much faster convergence. We also see how the VAEs tend to have a large KL divergence between the true posterior and the variational estimate due to this gap, we depict below samples from those models.

**Wave**

We propose here a simple example where the read data is as follows:

Figure 9: sample of noise data for the wave dataset

We train on this dataset the EM and VAE based learning with various learning rates and depict below the evolution of the NLL for all models, we also depict the samples after learning.

**MNIST** We now employ MNIST which consists of images of digits, and select the $4$ class. Note that due to complexity overhead we maintain a univariate latent space of the GDN and employ a three layer DGN with 8 and 16 hidden units. We provide first the evolution of the NLL through learning for all the training methods and then sample images from the trained DGNs demonstrating how for small DGNs EM learning is able to learn a better data distribution and thus generated realistic samples as opposed to VAEs which need much longer training steps.

Figure 11: Samples from the various models trained on the wave dataset. We can see on **top** the result of EM training where each column represents a different run, the remaining three rows correspond to the VAE training. Again, EM demonstrates much faster convergence, for VAE to reach the actual data distribution, much more updates are needed.

Figure 12: Evolution of the true data negative log-likelihood (in semilogy-y plot on MNIST (class 4) for EM and VAE training for a small DGN as described above. The experiments are repeated multiple times, we can see how the learning rate is clearly impacting the learning significantly despite the use of Adam, and that even with the large learning rate, the EM learning is able to reach lower NLL, in fact the quality of the generated samples of the EM modes is much higher as shows below.

Figure 13: Random samples from trained DGNs with EM or VAEs on a MNIST experiment (with digit 4). We see the ability of EM training to produce realistic and diversified samples despite using a latent space dimension of 1 and a small generative network.

## Footnotes

[2] looser condition can be put as the up-to-layer $\ell$ mapping is a CPA on a coarser partition than $\Omega$ but this is sufficient for our goal.

[3]https://pypi.org/project/pycddlib/

[4]https://inf.ethz.ch/personal/fukudak/cdd_home/index.html