[Reviews · NeurIPS 2020]

Review 1

Summary and Contributions: The paper focuses on learning parametric distributions with latent variables via EM method, where the decoder is piece-wise affine function. They propose to compute the posterior distribution explicitly, which makes E-step exact. For piece-wise affine decoder and Gaussian prior, latent space is partitioned into polytopes and authors enumerate all the polytopes and do marginalization over them explicitly. M-step is also performed analytically due to usage of Gaussian distributions for the decoder and prior. Authors illustrate effectiveness of proposed exact EM learning on toy examples and on MNIST, as compared to vanilla VAE.

Strengths: The fact that for small networks and low-dimensional latent space the authors were able to do EM learning exactly is interesting. The way authors solve the computational problem of marginalizing the latent variables is new to me.

Weaknesses: The main weakness is very limited applicability of the proposed method: it doesn't seem like a good fit for modern machine learning given the size limitations. Given this the claims of the paper are exaggerated. Aside from interesting computational approach the impact of the method is low. Empirical evaluation of the computational effort of the new algorithm is missing.

Correctness: The claims of the paper are exaggerated since the method only applies to small problems while the claims are general. Empirical methodology is missing analysis of computational cost.

Clarity: The paper is well-written, with extensive derivations.

Relation to Prior Work: Unlike the prior work that analyzed partitioning of latent space by piece-wise affine functions, the current work explicitly performs marginalization over it, this is implied in the text.

Reproducibility: Yes

Additional Feedback: - Given explicit marginalization of latent variables, why not minimize log-likelihood directly rather then using EM? I am probably missing something, so would appreciate a clarification of this. - Adding cost analysis of the algorithm would be helpful, as well as discussion of scaling it up to realistic problems. - in comparison to VAE in Sec 4.3 it is unclear if the authors compute log-likelihood or ELBO (both terms are used in a confusing way) - curves in most plots are not labelled *************post rebuttal*********************************** I'd like to thank the authors for their response and addressing the above points. I have updated my score. I am still not convinced of the usefulness of the approach beyond toy examples.


Review 2

Summary and Contributions: Deep generative models (DGMs), specifically variational autoencoders (VAEs), currently rely on variational inference and stochastic optimization of a lower bound to maximize likelihood since the analytic likelihood cannot be computed in general. This paper shows that in fact the likelihood can be computed analytically and maximized with analytic expectation maximization (EM) updates when the network uses affine piecewise nonlinearities like ReLU and leaky-ReLU. The key insight is that these networks induces a partition of the latent space that can be handled tractably when the prior and likelihood are both Gaussian. This paper analytically derives the posterior distribution, the marginal distribution, the expectation of the complete likelihood (for the E step), and the updates to the parameters (for the M step). These novel derivations allows the authors to perform EM on DGMs for the first time. Empirically, the paper shows that learning via EM produces generative models with higher likelihood and higher sample quality than maximizing the evidence lower bound (ELBO). ********************* Post rebuttal I thank the authors for their rebuttal. Please include the discussions that were promised in the rebuttal. I will be staying with my score of 8. ********************* *********************

Strengths: 1. This paper is the first to compute the likelihood and posterior distributions involved in the probabilistic model of a variational autoencoder, showing that they are mixtures of truncated Gaussians for deep models that use affine piecewise nonlinearities. Being able to compute these terms enables applications like model comparison 2. This paper is the first to analytically perform EM with variational autoencoders. This is desirable because EM is guaranteed to converge to a local optimum and usually requires fewer iterations than gradient-based optimization. 3. The paper's derivations enables for the first time a direct comparison of the true likelihood to the ELBO 4. The analytic form of the posterior derived in the paper strongly motivates using multimodal posterior approximations 5. The paper shows that despite using a 1 dimensional latent space with a 2 hidden layer neural network, training VAEs with EM on MNIST results in higher likelihood models with more diverse samples than maximizing the ELBO. Additionally EM-learning is shown to converge to a better optimum in fewer updates than variational inference for MNIST and 2 other toy datasets.

Weaknesses: # Weaknesses 1. The proposed algorithm to compute the terms required for EM is computationally intensive and scales poorly with the latent space dimensionality and model size. This limits the empirical evaluations to only small neural networks and 1 dimensional latent spaces. 2. The analytical computations are only possible when working with Gaussian priors and likelihoods for the VAE. It also may not be possible to analytically compute the terms derived in the paper when the nonlinearities are not affine. 3. The derivation requires assuming that the likelihood of the data conditioned on latent has a constant covariance with respect to the latent. However, this is not a huge problem because in practice, the covariance is often made constant for optimization stability.

Correctness: The claims and methods are correct.

Clarity: The paper is very well-written and clear. The authors did well to present their work in a mostly self contained way.

Relation to Prior Work: The authors did a very good job discussing prior work, including how their methods differ and relate to previous contributions.

Reproducibility: Yes

Additional Feedback: Line 38: "descend" -> "descent" Line 54: "akin" -> "akin to" Line 89: "circulent" -> "circulant" Line 89: ", and, " -> "and, " Line 100: "D_\ell" -> "D^\ell" Line 124: "Maximization" -> "Maximizing" Line 168: "p(t=r)" -> "Pr(t=r)" Line 201: "hte" -> "the" Line 207: "q^\ell" -> "q^\ell (z)" Line 246: "b ased" -> "based" Section 4.3: May want to explicitly mention that you also performed experiment on MNIST, but left it in the appendix q^{all} is sometimes used instead of q in the appendix Line 476: "speicific" -> "specific" Appendix E: Although you describe the algorithm in previous sections, you may also want to explain what "reduce", "SearchRegion", "flip" mean or reference the section where they are explained. Line 596: "we an" -> "with an" Line 599: "read" -> "real"


Review 3

Summary and Contributions: The authors propose an expectation-maximisation (EM) approach to learn deep latent variable models (DLVMs) with decoders using ReLU nonlinearities. To this end, they leverage the fact that ReLU networks are piecewise affine, leading to the DLVM being effectively a Gaussian mixture model akin to mixtures of probabilistic PCA (PPCA).

Strengths: The approach is quite fresh, and the paper is certainly thought-provoking, by taking an inference approach somewhat orthogonal to the mainstream one. The authors do not try to hide that the EM approach is still quite impractical in realistic settings.

Weaknesses: - As acknowledged by the authors, there are important computational limitations that prevent the EM approach from being use to train really powerful DLVMs. - Similar insight was present in a paper presented at last year's ICML, which limits the novelty and "thought-provokingness" of the paper (see "prior work" box). - I think that the experiments are quite limited, in particular VAEs with more expressive posteriors could be compared to as well (e.g. IWAE, normalising flow posterior).

Correctness: All theoretical claims look sound.

Clarity: The paper is extremely well written, and should be enjoyed by both neophytes and VAE aficionados.

Relation to Prior Work: Very similar insights were present in the following paper: Variational Laplace Autoencoders, Park et al., ICML 2019 In particular, Park et al. (2019) also leverage the link between ReLU-based decoders and PPCA. This makes some of the novelty claims of lines 52-53 wrong. A detailed discussion on the links between these two papers is really needed. Less importantly, a few additional papers on links between mixture models and deep latent variable models and traditional mixture models: Leveraging the Exact Likelihood of Deep Latent Variable Models, Mattei and Frellsen, NeurIPS 2019 Taming VAEs, Rezende and Viola, arXiv:1810.00597 Finally, the idea of using EM to train deep latent variable models was already present in Bishop et al. (1998), although in a very different form: The Generative Topographic Mapping, Bishop et al., Neural Computation (1998)

Reproducibility: Yes

Additional Feedback: Isn't Lemma 2 true even without the ReLU assumption? Your proof technique does not work in that case, but it seems to be true under probably weaker assumptions. ----- POST-REBUTTAL EDIT ----- I have read other reviews and the authors's rebuttal. I appreciate that the authors discussed the links with the ICML paper on Laplacian VAEs that I mentioned. I strongly encourage them to write a detailed discussion on this (possibly in the appendix) to the final version. Assuming that they will add such a discussion, I am willing to raise my score to 7. I also suggest that the authors add short discussions on the other papers I mentioned, in particular Bishop (1998).


Review 4

Summary and Contributions: The paper introduces a method for exactly computing the marginal, posterior and conditional distributions of a Deep Generative Network that uses Continuous Piece Affine (not that strong of an constraint). They also derive an Expectation Maximization algorithm, allowing gradient free training and which doesn't rely on a variational bound. This allows them to highlight the shortcomings of VAE training for DGNs . They use this algorithm to demonstrate some of the instability of the VAE training for a DGN is because of the variational approximation isn't sufficiently well capturing the posterior.

Strengths: Providing ground truth checking of methods is often a vital step in understanding what price you pay for using different approximations and approximate inference schemes. This makes it a relevant and potentially significant result for the community that will potentially inspire future work and improvements. These improvements could both be to to their method in turns, probably trading off computation speed in exchange for accuracy, and potential improvements to the variational inference from better understand its short-comings.

Weaknesses: The principal weakness is that the method is very impractical for applying for higher dimensional problems as the scaling will be very poor. This is not explored nor is the scaling discussed in any detail apart from a very briefly qualitative mention in the appendix.

Correctness: The paper appears methodologically sound and correct. The experiments are very limited and synthetic, but they do clearly demonstrate their method working and the short-comings of AVI when applied to the same problems. Figure 4 clearly shows that EM training ends up with a superior NLL to the VAE.

Clarity: Paper is clear and well written. It clearly goes explains each concept in turn. The figures are useful for understanding the different parts of the model. Minor corrections Line 201 - typo

Relation to Prior Work: It appears well cited and how it differs from previous work is clearly explained.

Reproducibility: Yes

Additional Feedback: While computational complexity is not the main aim of the paper, it would be informative to provide the number of regions found in the posterior for the synthetic examples used in the paper, and the computational time required for their method and the VAE approximation. -- Post rebuttal: Thank you to the authors for adding citations for to address the scalability of the method with number of layer/topologies and adding computation times for each step to the appendix. Additionally, approximate estimates of how quickly the number of regions grows in a "real" network would help in evaluating the network.

[Author Response · NeurIPS 2020]

*Reviewer #1:* We thank the reviewer for his/her review and suggestions. **Practicality and claims:** In term of practical use of the EM training we agree and explicitly acknowledged that it is computationally demanding. However, the obtained analytical results hold for any depth/with and nonlinearities (as long as they are piecewise affine); the results of the paper are thus general and can be used to gain in depth theoretical understanding of generative networks and their learning dynamics (from the explicit M step). The obtained analytical forms allow (i) to better design VAEs (now knowing the target posterior that variational inference approximates), (ii) to guide the design of variational distributions (for example favoring full covariance and multimodal posterior) as well as (iii) interpreting the learned parameters from the M-step. Those insights are gained despite the computational limits of the practical EM learning as they rely on the analytical derivations only. We will add further analysis and discussions on this. On that note, we also believe that tremendous future work can also be done to derive faster EM learning leveraging the obtained formula either by providing principled approximations of the per region Gaussian integrals or by approximation of the partitions; we believe that this paper is only the beginning of such research directions. **Direct log-likelihood maximization:** As in any missing variable model (here $z$ is unobserved, only $x$ is observed) one can not directly minimize the negative log likelihood and must first infer $z$. EM is one common strategy to do so based on the posterior $p(z|x)$; once $z$ is inferred, one can then do the maximization of the (now estimated) log-likelihood. **Adding cost analysis of the algorithm:** We will add in the appendix exact computation times and further details for each of the experiment and different architectures for the EM learning as well as each step involved (partition finding, region triangulation, per region integration). **Comparison to VAE and figures:** In Fig. 4 we compared the negative log-likelihood of both models. While a VAE can be trained using the standard variational inference strategy, we evaluate its NLL after training and compare with the generative deep network trained with EM. We will explicit this in the caption. We will also make the legend and labels clearer in the figures.

*Reviewer #2:* We thank the reviewer for their appreciation of the paper. We will correct the typos and explicit the pseudo-code as well as providing exact link with the implementation. **Computational limitations:** Indeed, the current analytical EM learning is computationally demanding, we believe that future work can be done on this point by (1) providing analytical form of gaussian integration on a convex polytope (this would remove the need of triangulation and then inclusion-exclusion formula) or by (2) providing principled approximation of those integrals. Note that our main contributions are the analytical derivations of the probability distributions and EM formula, the practical EM learning demonstrates the usefulness of those derivations. **Gaussian prior and piecewise affine nonlinearities:** The review is correct; this only applies to Gaussian prior and output distributions and with DN employing spline operators like ReLU, leaky-ReLU, abs. value, ... which includes a large part of current generative network architectures. Also, the proposed method (with exact partition and per region derivation) can be employed to different distributions as long as they are conjugate priors. We will add this note in the paper. **Constant covariance:** Indeed, this case covers the practical cases of training in current generative models, however more general cases could be considered and even different distributions. We believe that the proposed methodology (per region derivation) provides a general framework and as long as the prior and output distributions are conjugate priors, analytical forms should be obtainable. We will add this discussion in the paper.

*Reviewer #3:* we thank the reviewer for their careful review and appreciation of the paper. **Previous work:** We thank the reviewer for this relevant reference (which we denote by ICML2019 thereafter). ICML2019 relates linear VAEs to PPCA and propose a mode approximation of the posterior in turn producing a novel type of VAEs (Laplacian VAEs). ICML2019 also provides insights into the manifold geometry (piecewise affine) of ReLU VAEs. We will add this reference and detailed review in the background section. However we believe that none of our contributions is over-shadowed by ICML2019 since: (i) we extend the PPCA link of linear VAES to nonlinear VAEs resulting in MPPCA; (ii) we extend their geometrical insights to piecewise affine nonlinearities (not only ReLU) which consequently also allow to apply ICML2019 approximation methods to a broader class of VAEs; (iii) in ICML2019, no analytical (explicit) form is given for the probability distributions of a nonlinear VAE as the motivation of the paper was to provide a mode approximation based on a linearization of the network to tackle large scale tasks. We will also discuss the paper approximation method in the future work section as such posterior mode estimation could be employed and potentially improved with the proposed distributions. **Lemma 2 ReLU assumption:** you are correct, Lemma 2 holds for more general DGNs (as long as there is no surjectivity), we will add this note and discuss such cases in the paper.

*Reviewer #4:* We thank the reviewer for his/her appreciation of the paper and we agree that providing exact methods even with demanding computational cost is crucial to exactly measure the impact of current approximation methods in VAEs. **Computational complexity discussions:** indeed, the computational bottleneck comes from the number of regions that then need to be triangulated. We will add computational time of each of the involved steps in the appendix: (i) computation of the partition, (ii) triangulation of each region (on average) and (iii) integration on a region. We will provide those statistics for the few different topologies that were used in the paper. Concerning the rate of growth of the number of regions in a real network, we will add citations to the following papers: "Complexity of Linear Regions in Deep Networks", "On the Number of Linear Regions of Deep Neural Networks" and "A Spline Theory of Deep Networks" with discussions.

[Meta-Review · NeurIPS 2020]

The reviewers highlight a solid theoretical contribution that is interesting for the wider NeuRIPS community. The paper is accepted.